

# 1 Evaluation and Application of Precipitable Water Vapor Product
# 2 from MERSI-II onboard the Fengyun-3D Satellite

Wengang Zhang[1†], Ling Wang[2,3], Yang Yu[1], Guirong Xu[1], Xiuqing Hu[2,3], Zhikang Fu[1], Chunguang Cui[1]
[1]Hubei Key Laboratory for Heavy Rain Monitoring and Warning Research, Institute of Heavy Rain, China Meteorological
Administration, Wuhan 430205, China
[2]Key Laboratory of Radiometric Calibration and Validation for Environmental Satellites, China Meteorological
Administration, Beijing 100081, China
[3]National Satellite Meteorological Center, China Meteorological Administration, Beijing 100081, China
Correspondence to: Wengang Zhang (wengang812@whihr.com.cn)
**Abstract.** The evaluation of precipitable water vapor (PWV) derived from the advanced Medium Resolution Spectral Imager
(MERSI-II) onboard FengYun-3D is performed with the PWV from Integrated Global Radiosonde Archive (IGRA) based on
626 sites (54214 match-ups) in total during 2018-2021. The averaged PWVs from MERSI-II and IGRA both present the
distribution opposite to latitude, with great PWV mostly found in the tropics. In general, a good consistency exists between
the PWVs of MERSI-II and IGRA, and their correlation coefficient is 0.9400 and root mean squared error (RMSE) is 0.31
cm. The peak values of mean bias (MB) and the mean relative bias (MRB) are 0.00 cm and -2.38%, with the standard
deviations of 0.25 cm and 16.8%, respectively. For most sites, the PWV is underestimated with the MB between -0.28 cm
and 0.05 cm. However, there is also overestimated PWV, which is mostly distributed in the surrounding areas of the Black
Sea and the middle of South America. The peak values of MB are found in February and July over the Southern and
Northern Hemisphere, respectively. More than 66.91% of retrievals falling within the except error (EE) envelope during all
months. Overall, the MRB and RMSE become larger with the increasing temporal and distance discrepancy, and it is
contrast for EE and correlation coefficient. Besides, the distance discrepancy impacts the evaluation more. The application of
PWV product over Qinghai-Tibet Plateau shows that the transport of water vapor along the Brahmaputra Grand Canyon is
obvious and it is more significant in July.
**1 Introduction**
Water vapor is an important part of the atmosphere and widely known as the most important greenhouse gas and it can
significantly affect climate change, radiation balance and the hydrological cycle (Kiehl & Trenberth, 1997; Held & Soden,
2000; Dessler & Wong, 2009; Zhao et al., 2012). The spatiotemporal variations of water vapor are essential for
understanding formations of clouds and mesoscale meteorological systems in that cloud and precipitation always rely on the
changes of water vapor (Trenberth et al., 2003). Furthermore, water vapor can also influence the atmospheric transmittance





and upward radiance over the view of satellite. Therefore, the information of water vapor is highly required to correct
atmospheric effects in the satellite-based retrieval algorithm for land surface temperature (Meng et al., 2017).
Considering the critical role of water vapor, technologies aiming at the measurement of atmospheric water vapor have
been developed. The precipitable water vapor (PWV), which means the integrated water vapor contained in a vertical
column of a cross-sectional area, is an important indicator for the total atmospheric water vapor condition. The two major
methods used for measuring PWV are satellite-based and ground-based technologies. Several ground-based measurements,
such as radiosonde(Durre et al., 2009), global position system (GPS) receivers (Bevis et al., 1992), microwave radiometer
(MWR) (Westwater, 1978) and sun photometer (Alexandrovet al., 2009), have been deployed to monitor the variability of
water vapor. However, the spatial coverage of ground-based measurements is limited and inhomogeneous, and it is difficult
to obtain a wide range of observation from multiple sources to support the study for the distribution of PWV in both regional
and global scales. This is because the uncertainties in different measurements are not completely consistent, and they have
distinct discrepancies, even in the magnitudes (Chen & Liu, 2016; Wang et al., 2016). Different from the ground-based
measurements, the satellite-based measurement is more useful for the temporal analysis of PWV over a wide area. Especially,
the polar orbiting satellite-based measurements of water vapor have the considerable advantage due to their global coverage
with satisfactory temporal and spatial resolutions. Therefore, the polar orbiting satellite-based PWV product is widely used
for understanding the global distribution of water vapor. As we all know, the well knowledge of global water vapor
distributions is especially important for global atmospheric models aiming to predict weather or climate. Thus, the water
vapor products retrieved via polar orbiting satellite have become essential input parameters to sustain numerical models of
the atmosphere, especially where global water vapor information is required within a short time span, and the assimilation of
PWV has been proved that it can help improve precipitation forecasts (Rakesh et al., 2009).
There are three major satellite-borne sensors that can provide the global near-infrared (NIR) PWV products. The
Moderate Resolution Imaging Spectroradiometer (MODIS) onboard the Terra and Aqua polar orbiting satellite platforms is
one of the most important instruments for obtaining global PWV, and it has been widely used for a few decades since the
launching of Terra spacecraft in 1999. The Medium Resolution Imaging Spectrometer (MERIS) is one of ten instruments
built in Envisat, which was launched on 1 March 2002, but the mission was terminated on 8 April 2012 because of the loss
of contact with the satellite. For Chinese FengYun 3 (FY-3) meteorological series satellite, one of the major payloads
onboard is the Medium Resolution Spectral Imager (MERSI), which primarily monitors the ocean, land, atmosphere, etc.
FY-3D is the Chinese second-generation polar-orbiting meteorological satellite, equipped with the advanced MERSI
(MERSI-II), and it is launched on 15 November 2017. For MERIS, the PWV retrieval algorithm employing the ratio of top
of atmosphere (TOA) radiance at one water vapor absorption channel (900 nm) to TOA radiance at 885 nm, which is outside
water vapor absorption region (Bennartz and Fischer, 2001). However, both the algorithms for NIR PWV derivation of
MODIS and MERSI-II adopt the ratios of reflected solar radiance between three NIR water vapor absorption channels and



two non-absorption channels (Gao and Kaufman, 2003; Wang et al., 2021). The setup of non-absorption channels of
MERSI-II is same as that of MERSI but the absorption channels of MERSI-II are similar with those of MODIS. Besides, the
prelaunch and orbital calibration and characterization of MERSI-II were conducted to ensure the quality of its products (Xu
et al., 2018).
It is strongly necessary to evaluate the satellite-based PWV product ahead of its application in atmospheric science
research. The PWV from MODIS has been extensively evaluated through comparing it with the PWV derived from other
measurements. The GPS PWV is widely used for the evaluation of PWV derived from MODIS (Liu et al., 2006; Prasad and
Singh, 2009; Lu et al., 2011). Ground-based MWR, which can measure integrated water vapor with high temporal resolution
and has a reliable measurement under clear sky condition, is also used for the evaluation of MERIS PWV (Li et al., 2003).
Additionally, the radiosonde PWV, calculated from the integration of specific humidity, has been recognized to be a useful
benchmark, being used in evaluating the MODIS PWV in China (Liu et al., 2015), the Iberian Peninsula (Sobrino et al., 2014)
and Hong Kong (Liu et al., 2013). However, few studies have focused on the evaluation of the MERSI-II PWV up to now,
and the lack of effective assessments greatly limits the application of the MERSI-II PWV product, because the accuracy of
the product cannot be fully acknowledged.
Integrated Global Radiosonde Archive (IGRA) is the greatest and most comprehensive collection dataset of historical
and near real-time global quality-assured radiosonde observations. It has been used extensively in a variety of studies,
including model verification, atmospheric processes, and climate research. Moreover, the radiosonde PWV is also widely
applied in the assessments of measurements from other platforms, especially satellite derived PWV around the world
(Adeyemi and Schulz, 2012; Antón et al., 2015; Niilo et al., 2016). Consequently, the IGRA data are selected for the
evaluation of the PWV derived from MERSI-II in this study.
The purpose of this paper is to evaluate the MERSI-II PWV globally by comparing it with the global IGRA
observations. We are trying to explore the global performance of FY-3D MERSI-II PWV and analyzing the influence factors
in the evaluation. Besides, the application of MERSI-II PWV on the study for the distribution of PWV over Qinghai-Tibet
Plateau (QTP) is also discussed. The structure of this paper is arranged as follows: Data sources and details are discussed in
Section 2. Section 3 presents the merging procedures methodology applied in the sample selection. The evaluation results of
MERSI-II PWV against the PWV from IGRA are presented in Section 4. A discussion and conclusion of the forementioned
results are given in the final section.
**2 Data description**
The satellite-based PWV product used in this paper is derived from FY-3D MERSI-II, and the ground-based
measurements are the AERONET and IGRA derived PWV data.



## 2.1 MERSI-II PWV

FY-3D, which was successfully launched on 15 November 2017, is the fourth and latest satellite of Chinese second-generation polar-orbiting meteorological satellite. It is operated in a sun-synchronous orbit at an average altitude of 830.73 km, passing over the equator at 13:40 local time (Yang et al., 2019). The MERSI is one of the major instruments carried by FY-3 series satellites, and the MERSI-II onboard FY-3D is an upgraded version of the first-generation instrument. A series of comprehensive prelaunch calibration have been operated to ensure the high quality of the products from MERSI-II (Xu et al., 2018), which was from MERSI and has been significantly improved with high-precision on-board calibration and lunar calibration capabilities (Wu et al., 2020). Besides, MERSI-II has 25 channels with a spectral coverage from 0.412 μm to 12.0 μm, and the NIR PWV products of FY-3D are retrieved with three absorption channels (bands 16, 17 and 18) and two non-absorption channels (bands 15 and 19) in the 0.8-1.3 μm range with a spatial resolution of 1 km at nadir (Wang et al., 2021). The water vapor absorption channels of MERSI-II, which is now similar with those of MODIS, are reselected because the three absorption bands have different sensitivities to various water vapor conditions. Therefore, MERSI-II is more useful in the retrieval of water vapor under different conditions (dry, medium, and humid). The NIR PWV product derived from MERSI-II can be accessed on the website of http://satellite.nsmc.org.cn/PortalSite/Data/Satellite.aspx.As we all know, the near-infrared precipitable water vapor product from MERSI-II is the total column amount of water vapor over cloudless land of the globe as well as above clouds. Besides, over the oceanic areas with sun glint the PWV product can also be obtained. However, in order to consist with the ground-based measurements, only the PWV product over cloudless land area is used in this study. The data span is from September 2018 to June 2021 with a spatial resolution of 1 km×1 km.

## 2.2 Radiosonde

Integrated Global Radiosonde Archive (IGRA) which is a collection of historical and near real-time global radiosonde observations, is archived and distributed by the National Centers for Environmental Information (NCEI), formerly the National Climatic Data Center (NCDC), and it can be accessed at ftp://ftp.ncdc.noaa.gov/pub/data/igra. Version 2 of IGRA (IGRA 2) is used in this study. A total of 33 data sources, including 10 out of 11 source datasets used in IGRA 1, have been integrated into IGRA 2, which was fully operational on August 15, 2016 and has a higher spatial and temporal coverage. Therefore, compared to IGRA 1, the IGRA 2 contains nearly twice as many sounding stations and 30% more soundings. Sounding-derived parameters are recorded according to separated station files and continue to be updated daily, and PWV is one of the derived parameters. PWV will be calculated if the pressure, temperature, and dew point depression are available from surface to the level of 500 hPa (Durre et al., 2009). The calculation involves the acquirement of specific humidity at each observation level and then the integration of specific humidity between the surface and the level of 500 hPa, so IGRA-



derived PWV is recognized as surface-to-500-hPa PWV. Due to the time range of IGRA data, there are only 625 out of 1535
global IGRA stations can be matched with the FY-3D MERSI-II PWV products.

**2.3 AERONET**

The federated Aerosol Robotic Network (AERONET) is a network of ground-based Cimel Electronique Sun
photometry, which can measure beam irradiance and directional sky radiance routinely during the daytime in clear
conditions (Holben et al., 1998). AERONET was established by NASA and PHOTONS (PHOtométrie pour le Traitement
Opérationnel de Normalisation Satellitaire), primarily aiming to provide public domain dataset of global aerosol optical and
microphysical properties. In addition, based on the measurements at the 940 nm water-vapor channel and the atmospheric
window band centered at 870 nm and 1020 nm, PWV was also calculated (Che et al., 2016). The AERONET version 3
database provides three levels of data: Level 1.0 (unscreened), Level 1.5 (cloud-screened), and Level 2.0 (cloud-screened
and quality-assured), and it can be accessed at https://aeronet.gsfc.nasa.gov. Level 2.0 dataset, which is used in this study,
signifies an automatically cloud-cleared, manually quality-controlled dataset with pre- and post-field calibrations applied.
All the instruments in the AERONET are annually calibrated with reference to the world standard: the Mauna Loa
Observatory (Malderen et al., 2014). Thus, the measuring accuracies of different AERONET stations are accurate and
consistent (Liu et al., 2013). As discussed by Pérez-Ramírez et al. (2014), PWV obtained from AERONET has a dry bias of
approximately 0.16 cm against radiosonde PWV and it is reasonable for the meteorological studies.

**3 Methodology**

**3.1 Statistical indicators**

The common statistical indicators, such as the mean bias (MB), the mean relative bias (MRB), correlation coefficient
(CC) and the root mean squared error (RMSE), are used to evaluate the precision of the retrieved PWV from MERSI-II. The
MB, which can indicate the tendency of underestimation or overestimation, is desirable to be close to zero. The MRB can be
defined as the percentage deviation between the derived and observed PWVs, and its perfect value is 0. CC is an indicator
that can quantify the agreement between PWVs of MERSI-II and IGRA, and the closer of the CC to 1 means a better
coherence. The RMSE reflects the actual deviations between the paired derived value and reference value, and lower RMSE
values are preferred with a perfect value of 0. Besides, we adopt the percentage of matching data falling within an expected
error (EE) envelope and it is expressed as EE value in this paper. EE envelope is popularly used in the evaluation of aerosol
retrievals, and an EE value of >66% indicates satisfactory agreement (Levy et al., 2010). In addition, the EE envelope
defined as ±15% is also used in the validation analysis of MODIS derived PWV product the same (Martins et al., 2019). All
the calculations of indicators are presented as follows:



$$MB = \frac{1}{N}\sum_{i=1}^{N}(PWV_{si} - PWV_{gi}) \ ,$$ (1)
$$MRB = \frac{1}{N}\sum_{i=1}^{N}(\frac{PWV_{si} - PWV_{gi}}{PWV_{gi}}) \times 100\% \ ,$$ (2)
$$CC = \frac{\sum_{i=1}^{N}(PWV_{si} - \overline{PWV_{si}})(PWV_{gi} - \overline{PWV_{gi}})}{\sqrt{\sum_{i=1}^{N}(PWV_{si} - \overline{PWV_{si}})^2 \sum_{i=1}^{N}(PWV_{gi} - \overline{PWV_{gi}})}} \ ,$$ (3)
$$RMSE = \sqrt{\frac{1}{N}\sum_{i=1}^{N}(PWV_{si} - PWV_{gi})^2}$$ (4)
$$EE = \pm(0.05 + 0.15 \times PWV_g)$$ (5)
where $PWV_s$ is the MERSI-II PWV product, $PWV_g$ is the IGRA PWV product, and N is the total number of match-up.
**3.2 Collocation strategy**
As we have mentioned above, FY-3D is primary operated in a sun-synchronous orbit with an equator crossing time at
13:40 local time. However, radiosonde is released at 00:00 UTC and 12:00 UTC and there is a significant temporal
discrepancy between satellite and radiosonde at most sites. Besides, the distribution of radiosonde site is sparse over globe.
For the evaluation of PWV from global reanalysis models with a temporal resolution of 6 h, temporal window of ± 3 h and
distance of ± 50 km is employed in the comparison with PWV from Maritime Aerosol Network (Pérez-Ramírez et al., 2019).
In order to determine the temporal collocation window that can adequately match the satellite with the ground-based
measurements, the consistency between the existing AERONET PWV and AERONET PWV measurements in various
temporal discrepancy intervals from 1 h to 6 h is analyzed. In processing, only the point that has matching data in each
interval is selected for the comparison reliability. The results are presented in Figure 1, and obviously, there is a good
consistency at all situations with the CC larger than 0.9690 and the slope is larger than 0.965. Although MRB and RMSE
become larger with the increasing temporal interval, their values are less than 1.70% and 0.23 cm, respectively. Moreover, it
can be observed that the MB values of all comparisons are 0.00 cm, which suggests that the bias is distributed equally
around zero. More than 80.92% match points are within the EE and the largest value is 99.97% when the temporal
discrepancy is within 1 h. Therefore, we make the conclusion that the temporal collocation window for PWV evaluation can
be set to 6 h.



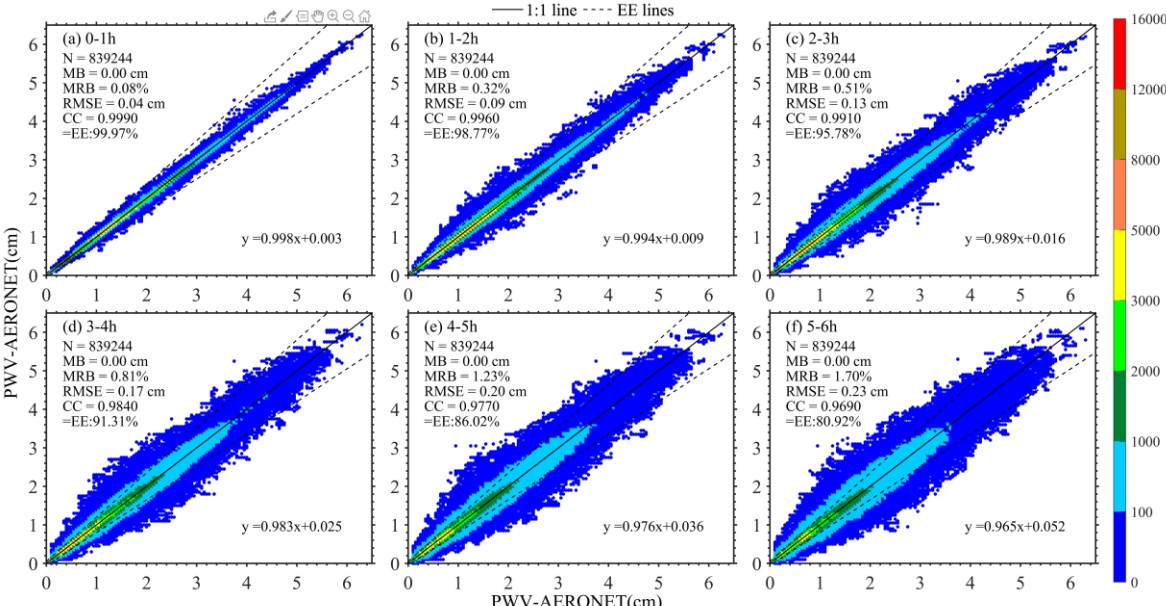


Figure 1 Scatter plots of PWV derived from AERONET in different temporal discrepancy intervals and (a)-(f) present the temporal discrepancy of 0-1 h, 1-2 h, 2-3 h, 3-4 h, 4-5 h and 5-6 h, respectively. The solid line represents the line 1:1, dashed lines are the envelope lines of EE. The color bar depicts the number density of match-ups for each bin of PWV in a 0.01 cm×0.01 cm grid. Proportion of matching data falling into EE envelope is presented by =EE value.

For the MERSI-II, the spatial resolution at nadir is 1 km × 1 km for NIR bands, which are used for the retrieval of PWV. Therefore, we use the standard deviation (STD) of a box with 9×9 pixels to eliminate the invalid PWV measurement. In operation, we set a general principle that the STD of this selected box must be less than 0.25 cm and the value of the STD dividing the minimum within the selected box must be less than 1. Otherwise, the data is marked as unreliable and will not be selected for the comparison. This is because the PWV in clear sky is considered as less varied in a local area based on the analysis of PWV derived from AERONET. In addition, the cloud mask (CLM) product of MERSI-II is applied in the collection of comparison samples of MERSI-II PWV and radiosonde PWV. For the MERSI-II CLM product, there are four clear-sky confidence levels (confidently clear, probably clear, probably cloudy, cloudy) for each pixel and they are denoted by the values of 3, 2, 1, and 0, respectively. Only the situation in which all pixels of the selected box are confidently clear is considered and collected for MERSI-II PWV product. Unfortunately, there is no cloud measurement in radiosonde observation, so the cloud detection method with the relative humidity threshold of sounding is employed here (Zhang, 2010), and then the cloudless radiosonde PWV dataset is established.

In processing, all the PWV retrievals derived from MERSI-II within ±6 h of radiosonde release time are all collected and the closest PWV retrieval of MERSI-II within 100 km distanced from the IGRA site is selected and matched up with IGRA PWV. Figure 2 illustrates the available sample numbers of radiosonde sites over the globe during 2018-2021, with





totally 626 sites. The sample numbers of all sites vary from 15 to 474, and observations are concentrated in the Northern
Hemisphere. Around the equator, few samples are got due to the high occurrence frequency of clouds and precipitation.
Most frequently sampled places are China, Europe, and northern America, where IGRA sites are densely distributed, while
there are few match-ups in Africa because radiosonde stations associated with IGRA are rarely seen there.

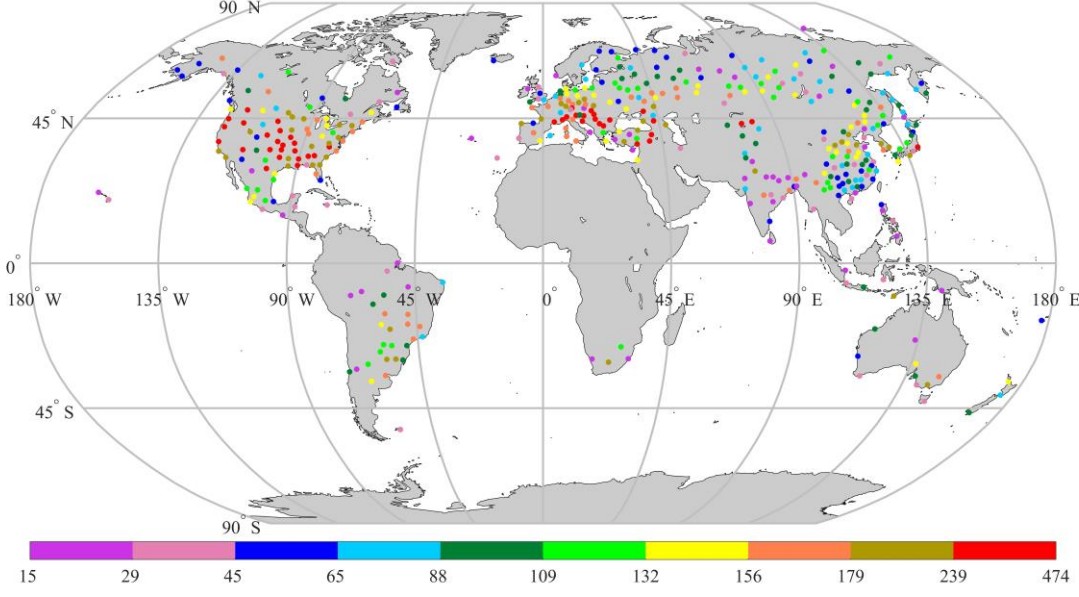


Figure 2 Number of matchups between MERSI-II and IGRA PWV observations for each site during 2018-2021.

## 4 Results and Discussion

### 4.1 Global comparison

Figure 3 illustrates the global averaged PWV obtained from the MERSI-II and IGRA under clear sky conditions. In
general, both the averaged PWV derived from MERSI-II and IGRA show the distribution opposite to latitude, with large
PWV values mostly found in the tropics but rare in high latitude. Around the tropics with latitude between 20°S and 20°N,
the greatest PWV values are found with most PWV values above 2.17 cm. Lower PWV values are presented in mid-latitude,
but the variability of PWV is the largest here with the values range from 0.60 cm to 2.17 cm. The PWV values in high
latitudes are the lowest and most sites have the PWV values below 1.44 cm. The global distribution of averaged PWV is
uneven and generally characterized by one low and two high PWV centers. The low center is in the east of Russia and the
northeast of China, with PWV below 1.16 cm measured at most sites. The two high centers are in the surrounding areas of
the Bay of Bengal and the middle part of South America, with most PWV values larger than 1.72 cm.


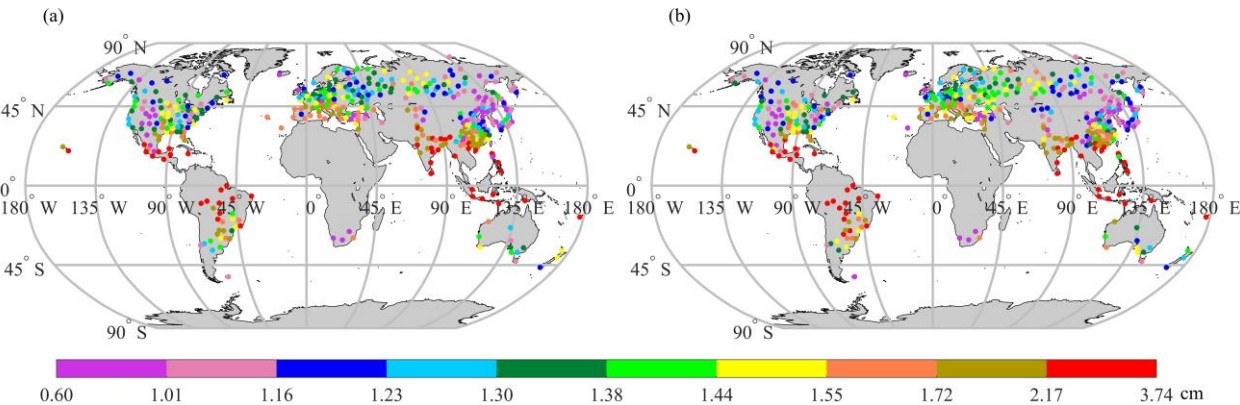


Figure 3 Global averages of PWV derived from MERSI-II (a) and IGRA (b) for each site.

Figure 4a shows the scatter plots of PWV derived from MERSI-II against IGRA observations. There are 54214 match-ups in total and the MERSI-II (IGRA) PWV ranges from 0.1(0.0) cm to 4.7 (5.9) cm, with a high number density between 0.2 cm and 2.0 cm. It is found that the MERSI-II and IGRA PWV measurements are well correlated with CC of 0.9400, while the retrieved PWV from MERSI-II is slightly underestimated, with MB of -0.09 cm and MRB of -1.90%. Besides, the RMSE is 0.31 cm and the EE value is satisfactory (75.36%), and the statistical biases are slight larger than those from the evaluation of MODIS over globe by comparing with the observations of AERONET (Martins et al., 2019). It is considerable that satellite has a larger temporal discrepancy with radiosonde than AERONET, which has a high temporal resolution about 1 min, and this will also cause the increasing error in the evaluation of MERSI-II PWV product. Although the reasonable MB and MRB have been found in the evaluation of all sites, there are some individual points with the unnormal MB and MRB. Therefore, the top 1% and bottom 1% of MB and MRB are not present in the histogram in order to show an intuitive acknowledge of distributions of MB and MRB. Figure 4b reveals the distribution of MB between FY-3D MERSI-II and IGRA, and notably, the MB is concentrated around zero and the bias distribution is left-skewed, which means that there are more negative MB values. However, the peak value of MB is 0.00 cm and there are 23.8% of all points within the interval from -0.05 cm to 0.05 cm, and the STD of MB is 0.25 cm. It can be concluded that there is a high accuracy for MERSI-II PWV product, as evidenced by the low MB and STD which are similar with those in the evaluation of ground-based GPS PWV against radiosonde PWV (Wang et al., 2007). For the MRB shown in Figure 4c, the distribution is also centered around zero but with a right-skewed pattern, and the peak value of MRB is -2.38% with the STD of 16.8%. The highest frequency of interval ranges from -4.0% to -2.0%, with more than 5.9% of all retrievals falling within this interval. And this result is comparable to the accuracy of MODIS NIR PWV product, which is compared with MWR PWV and with a 5%-10% error range (Gao & Kaufman, 2003). Besides, the analysis and explorations of high MB and MRB values indicate that the dominant large values of MB and MRB are caused by the matchups with high temporal discrepancy or large distance between FY-3D and IGRA observations.





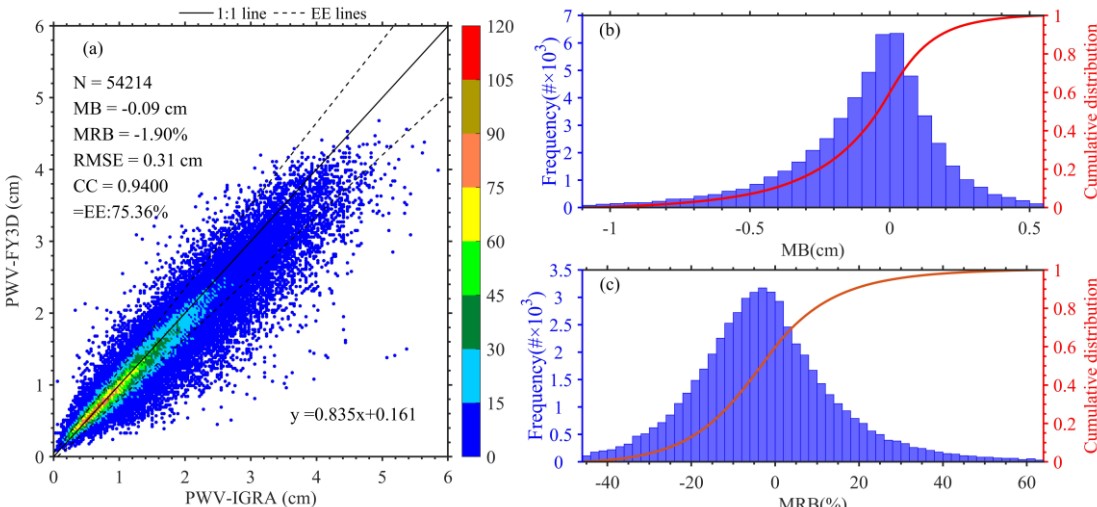

233

Figure 4 (a) Total density scatterplot of the PWV derived from MERSI-II against that of IGRA for all sites. Frequency

histograms of (b) MB and (c) MRB between MERSI-II and IGRA PWV superimposed on a cumulative distribution curve.

Figure 5 shows the geographical distributions of PWV comparison statistics between MERSI-II and IGRA over globe.

As we can see from the MB distribution in Figure 5a, the MB presents low values between -0.28 cm and 0.05 cm at 80% of

the sites. About 80% of all sites have negative MB values, and this indicates that PWVs derived from MERSI-II are

primarily underestimated compared with IGRA PWV values. There are 10% of all sites with larger MB values larger than -

0.28 cm, and most sites are distributed in the west and south of Asia. Those sites with overestimated PWV values of MERSI-

II are mostly distributed in the surrounding areas of the Black Sea and the central South America, and most of them have the

MB values larger than 0.05 cm. It is also found in the evaluation of PWV product derived from MODIS onboard Terra and

Aqua, and the MB of MERSI-II is slight smaller, especially compared with that of Terra (Martins et al., 2019). In general,

the distribution of the MRB (Figure 5b) is similar with that of the MB at most sites. However, there are two areas that have

slight discrepancies between them. One area is in eastern Russia and northeastern China, where there are some sites with the

larger MRB values above 4.45%, although the MBs are small over this aera with the values range from -0.08 cm to 0.05 cm.

As we can see from figure 3, there is a low averaged PWV value in this region, and this is the dominant reason for the great

MRB values but with small MB values over this aera. Another area is the middle part of South America, where the sites have

large MB values and comparatively low MRB values, and this is because the large mean PWV values in this region. The

larger evaluation error of PWVs derived from MODIS and reanalysis products also have been found in the middle of South

America, with most sites have the MB and RMSE both larger than 0.4 cm (Lu, 2019; Wang et al., 2020). Figure 4c depicts

the distributions of RMSE for all sites and RMSE present low values with 90% of sites below 0.49 cm. The large RMSE

values are primarily found at low latitudes, mostly in South and Southeast Asia. However, in the east of Europe, there are

small RMSE with values below 0.21 cm at most sites. In general, there is a good agreement between MERSI-II and IGRA





PWV at most sites with the CC value above 0.8782. The high correlated sites are mainly distributed around the east of
Europe and have the CC values larger than 0.9557, while the low CC values that smaller than 0.8213 are predominantly
concentrated around the equator.

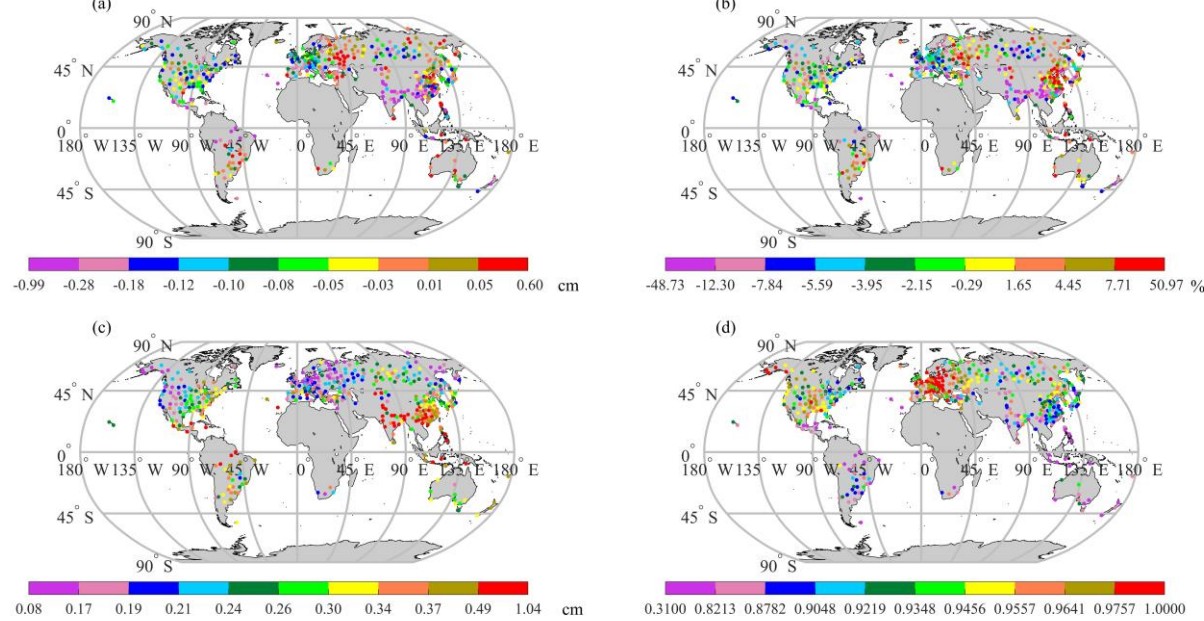


Figure 5 The geographical distributions of PWV comparison statistics between MERSI-II and IGRA.
**4.2 Temporal variations analysis**
As we have mentioned above, PWV presents a notable temporal variation, and the seasonal variation is a key point to
characterize the climatic change of PWV. Therefore, a seasonal comparison between the PWVs derived from MERSI-II and
IGRA is firstly analyzed in this section and the results are given in Figure 6. The four seasons in the Northern Hemisphere
are defined as follows: spring (MAM), summer (JJA), autumn (SON) and winter (DJF), and it is opposite in the Southern
Hemisphere. There are slight underestimations of MERSI-II derived PWV for all the four seasons. The MBs in summer and
autumn are both large with the magnitude greater than 0.09 cm but with negative values, and the MB value is larger in
summer. With abundant water vapor in summer, thin clouds easily form but they are hardly to be measured by satellite.
Therefore, the PWVs from MERSI-II are often underestimated due to the weakened or covered radiation signal under the
cloud. The RMSE is within 0.35 cm in all the four seasons, especially in winter when the RMSE is 0.27 cm and MB is at the
value of -0.04 cm. Besides, the MRB also presents the similar seasonal variations, with a peak value in summer and a
minimum value in spring. The MRB is positive in winter, and this may be related to the small PWVs with a high positive
MB in winter. Moreover, the PWV derived from MERSI-II has strong correlations with IGRA PWV and the CC is larger





than 0.9080 in all seasons. However, the EE values, ranging between 73.10% (winter) and 77.47% (summer), do not show
obvious seasonal variations.

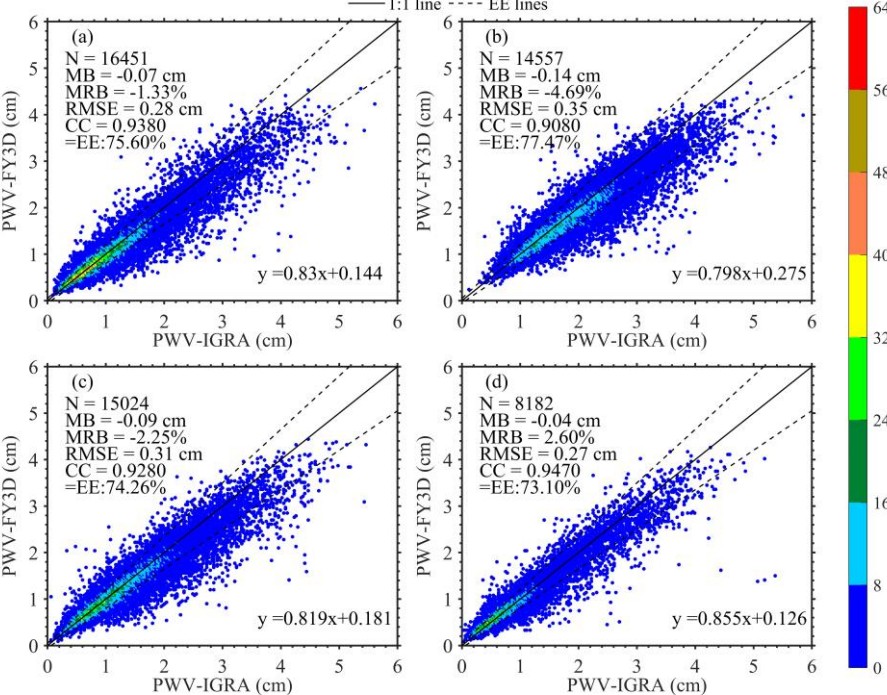


Figure 6 Seasonal scatterplots for the PWV comparison between MERSI-II and IGRA in (a) spring, (b) summer, (c) autumn
and (d) winter.

In addition, the monthly performance of the MERSI-II PWV product is also evaluated. Table 1 demonstrates the results

compared with PWV of IGRA. The number of match-ups ranges from 1847 (224) to 5956 (742) in the Northern (Southern)
Hemisphere. The MERSI-II PWV is underestimated in all months, and the underestimation is more significant in the
Northern Hemisphere. The slope values are less than 0.832, which is the fit-slope in March. However, the fit-slope in the
Southern Hemisphere is greater than 0.801 except in January and February, and the greatest and smallest values are 0.874
(June) and 0.754 (February). Most of MB values are less than 0.10 cm and the peak MB values mainly appear in February in
the Southern Hemisphere and July in the Northern Hemisphere. For MRB, the variation is within a large range, and the
largest MRBs are in June and July over the Southern and Northern Hemispheres, with values being 4.06% and -5.43%,
respectively. During warm seasons, MRB is negative in most cases, but positive in cold seasons. The RMSE in the Northern
Hemisphere is slightly smaller than that in the Southern Hemisphere, where the greatest RMSE value is 0.40 cm in
December. Besides, there is a better correlation between PWVs derived from MERSI-II and IGRA in the Northern



Hemisphere, and all CC values are larger than 0.9070 except in July. The percentages of within EE envelope lines are all
larger than 66%, which is the threshold value of satisfactory consistency.
Table 1 Monthly statistics of comparison between PWVs derived from MERSI-II and IGRA in the Northern (Southern)
Hemisphere

| Month | N | Slope | MB (cm) | MRB (%) | RMSE (cm) | CC | Within EE (%) |
|---|---|---|---|---|---|---|---|
| Jan | 1847(224) | 0.812(0.760) | -0.05(-0.06) | 2.49(-0.13) | 0.24(0.37) | 0.9430(0.8550) | 74.01(70.98) |
| Feb | 2008(230) | 0.818(0.754) | -0.06(-0.13) | 0.32(-3.77) | 0.23(0.38) | 0.9510(0.8910) | 75.30(72.61) |
| Mar | 2868(238) | 0.832(0.814) | -0.06(-0.08) | 0.70(-2.04) | 0.25(0.37) | 0.9510(0.8890) | 76.01(74.79) |
| Apr | 5956(369) | 0.802(0.808) | -0.06(-0.04) | -0.63(0.34) | 0.25(0.36) | 0.9410(0.8880) | 76.83(75.61) |
| May | 5903(468) | 0.786(0.872) | -0.10(-0.02) | -3.65(2.19) | 0.30(0.30) | 0.9170(0.9460) | 76.05(78.21) |
| Jun | 5796(516) | 0.802(0.874) | -0.12(0.01) | -4.34(4.06) | 0.32(0.26) | 0.9140(0.9620) | 78.33(80.23) |
| Jul | 3993(558) | 0.792(0.873) | -0.16(-0.04) | -5.43(2.01) | 0.37(0.31) | 0.8980(0.9420) | 77.13(77.78) |
| Aug | 3974(669) | 0.797(0.850) | -0.15(-0.02) | -5.14(3.10) | 0.37(0.34) | 0.9070(0.9400) | 77.81(72.94) |
| Sep | 5189(742) | 0.816(0.846) | -0.11(-0.05) | -3.77(0.35) | 0.32(0.36) | 0.9180(0.9270) | 76.41(71.29) |
| Oct | 5072(538) | 0.804(0.873) | -0.09(-0.02) | -2.39(1.89) | 0.31(0.39) | 0.9270(0.9180) | 74.17(66.91) |
| Nov | 3688(444) | 0.783(0.838) | -0.08(-0.06) | -0.77(0.28) | 0.30(0.35) | 0.9270(0.9120) | 70.69(68.24) |
| Dec | 2584(340) | 0.763(0.801) | -0.05(-0.06) | 4.15(-0.44) | 0.30(0.40) | 0.9120(0.8720) | 68.34(70.59) |

**4.3 The influence factors on evaluation**
In this section, the MERSI-II PWVs with different temporal and distance intervals are compared with the IGRA PWV
in order to explore the effects of dissimilar discrepancies of time and distance on the evaluation of MERSI-II PWV.
Furthermore, the altitude difference has an important influence on the accuracy evaluation of PWV, so we also present the
deviation of MERSI-II PWV for each class of station altitude. Besides, the statistics in different latitudes is presented for
analyzing the accuracy of MERSI-II PWV over different regions.
Firstly, the comparison results between the MERSI-II PWV and the IGRA PWV at different temporal intervals are
shown in Figure 7. The MRB has significant differences at different temporal intervals. For MRB, the largest value of -3.73%
appears under the condition with temporal discrepancy of 0-1 h, and the minimum value is -1.13% when the temporal
discrepancy is 1-2 h. Moreover, the EE value varies obviously from 68.04% to 88.82%, and the value decreases with the
increasing temporal discrepancy. RMSE changes from 0.23 cm to 0.36 cm with the increasing temporal discrepancy. PWVs
from MERSI-II in all situations are highly correlated to the IGRA PWV with the CC values larger than 0.9320 in general,
and they have the best correlation when the temporal discrepancy is less than 1 h. However, there is no noteworthy
difference in different temporal intervals for MB. The MB with the temporal discrepancy of 1-2 h gets to the minimum at the
value of -0.06 cm, and it is slightly different in other situations with values range from -0.07 cm to -0.10 cm. What's more,
the slope of fitted line indicates that there is an obvious underestimation in the retrievals of PWV from MERSI-II, and the




most underestimated PWV is in the condition of 4-6 h temporal discrepancy with the slope value of 0.826. When it is clear sky, there is a slight temporal variation of atmosphere water vapor, resulting in the unapparent differences at different temporal discrepancy intervals. Figure 7d shows the comparison with the temporal interval of 4-6 h. Obviously, there is a great number of points about half of all at this interval, and this is because the matchups are mostly located over East Asia and North America, where a temporal discrepancy of 4-6 h exists between the passing time of FY-3D and the release time of radiosonde.

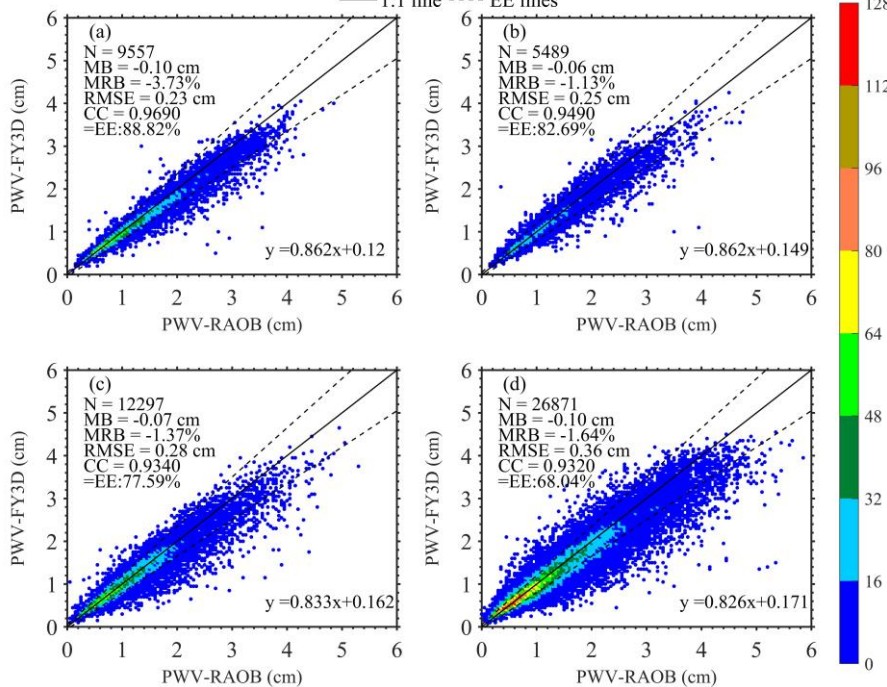

Figure 7 Scatterplots of the PWV comparison between MERSI-II and IGRA at temporal intervals of (a) 0-1 h, (b)1-2 h, (c) 2-4 h and (d) 4-6 h.

Figure 8 presents the results of comparison between the MERSI-II PWV and the IGRA PWV in different distance intervals. Most points are located within the distance interval of 0-5 km, and the number of points is 28756 out of all 54214 points. The MB increases with the extension of the distance between IGRA station and the footprint of MERSI-II, and the largest MB is -0.15 cm when the distance is within the range of 20-100 km. For the MRB, a more obvious difference is present as the value increases from -0.49% to -5.93% with the increasing distance. In all distance intervals, the RMSE has a satisfied value within the range of 0.28-0.40 cm. The large RMSE in the distance condition of 20-100 km is mainly caused by the obvious underestimation of MERSI-II PWV at some points. Overall, a good correlation exists between MERSI-II PWV and IGRA PWV with the CC value larger than 0.9060, which is less than that in the effect of temporal discrepancy on

evaluation. Besides, there are larger MB, MRB and RMSE in the evaluation of MERSI-II PWV with different distance
discrepancy intervals. Consequently, the discrepancy of distance is a more influential factor than temporal discrepancy on
the evaluation of PWV. Most points are located within the EE, and the EE value gets to decrease with the rise in distance and
the value is all above 68.58%.

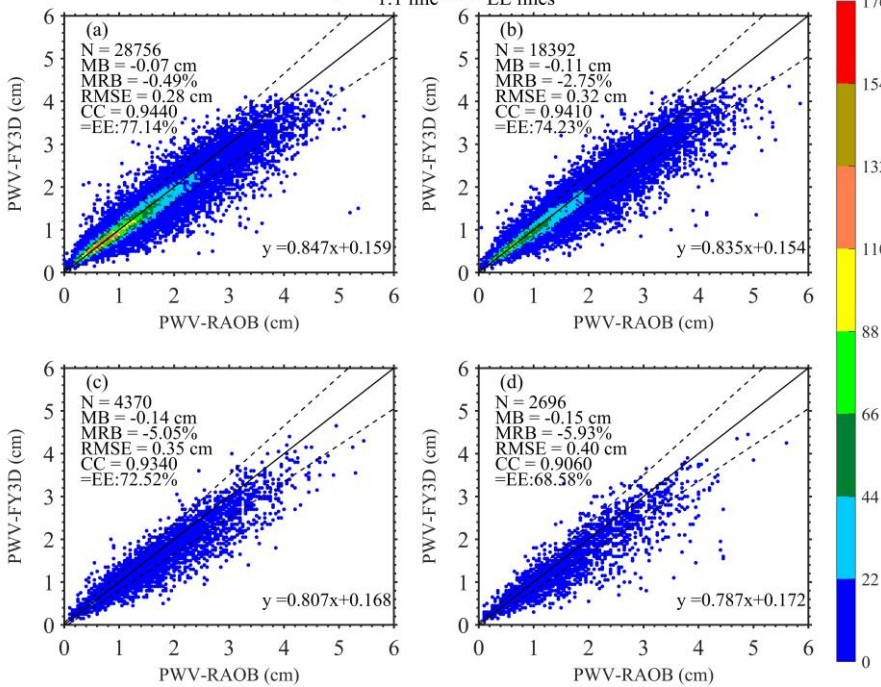


Figure 8 Scatterplots of the PWV comparison between MERSI-II and IGRA at the distance intervals of (a) 0-5 km, (b) 5-10
km, (c)10-20 km and (d) 20-100 km.

Table 2 illustrates the comparison results of the MERSI-II PWV in different intervals of altitude and latitude. Note that

only the observations in April are selected in the comparison rather than the annual mean value (MEAN), and this is because
averaging will smooth out the influence of altitude and latitude, which is highly related with the local climate situation, on
the PWV retrievals. First, most observations are collected at low altitude below 200 m, and the MEAN of PWV is largest in
the low altitude. The STD becomes smaller with the increase of altitude, which indicates that the PWV tends to be stabilized.
There is a small slope of linear fit in the high altitude, so we make the conclusion that MERSI-II PWV has an
underestimation, and the MB value alters from -0.08 cm to -0.02 cm for all altitudes. The largest MB appears in the sites
with high altitude, and RMSE also has the largest value of 0.28 cm, and it is also found in the evaluation of AIRS PWV (Qin
et al., 2012). This is because PWV is highly dependent on the altitude (Jiang et al., 2019), however, there is no height
correction that can be used to eliminate false signals especially over complex terrain during the processing of MERSI-II





PWV. The height correction is used in the validation of HY-2A PWV product and it is proved that can significantly reduce
the RMSE from 0.50 cm to 0.21 cm, especially for the sites over 1000 m.
In addition, the EE value ranges from 71.04% to 79.42% at all altitudes, and it is least in high altitude sites. There are
also larger MB and RMSE in the low altitude below 100 m, with the values are -0.07 cm and 0.28 cm, respectively. In the
hazy conditions with high humidity over the low altitude sites, the uncertainty in the amount of haze is one of the largest
error sources in the retrievals of PWV (Gao and Kaufman, 2003), however, the influence of haze is hardly corrected
completely in the MERSI-II PWV retrieval algorithm. There is a high correlation between MERSI-II PWV and IGRA PWV,
and the CC value is all above 0.8950. and the comparison of altitudes within 100-200 m presents a better performance.
The latitudinal distribution of PWV plays a key role in the study on the climatic change of global water vapor.
Consequently, the latitude is divided in a step length of 15-20 degrees to analyze latitudinal performance of MERSI-II PWV
product. Most of its samples are distributed from 20 °N to 50 °N, with the number of match-ups being totally 4249. The
MEAN of PWV presents an obviously distribution opposite to latitude, with the largest value of 2.94 cm within 0°N -20°N.
Meanwhile, the STD value in this region is also the largest, with the value of 1.01 cm. There also exists underestimation to
the fit-slope value ranging from 0.716 to 0.860. Furthermore, the MB and MRB values are mostly negative as well, and the
largest values of MB and MRB are within 0°N -20°N and 20°N -35°N, respectively. MERSI-II PWV has a great accuracy
over high latitudes of the Southern Hemisphere, and the RMSE is less than 0.19 cm above the latitude of 35 °S. However,
around the equator, the RMSE is large with the value greater than 0.43 cm. As discussed by Alraddawi et al (2018), for
MODIS PWV, there are also noteworthy latitudinal decreases in MB, MRB and RMSE. With abundant water vapor around
the equator, the cloud is easily formed and can micrify the PWV derived from MERSI-II because the MERSI-II can only
measure the conditions above clouds. In addition, the temporal discrepancy can also lead to the bias because the discrepancy
in the equatorial region is slight larger than in other regions overall.
Table 2 Summary statistics of MERSI-II PWV retrievals for different altitudes and latitudes in April.

| | Intervals | N | MEAN (cm) | STD (cm) | Slope | MB (cm) | MRB (%) | RMSE (cm) | CC | Within EE (%) |
|---|---|---|---|---|---|---|---|---|---|---|
| Altitude (m) | [-50 100] | 2593 | 1.24 | 0.79 | 0.828 | -0.07 | -1.25 | 0.28 | 0.9400 | 76.48 |
| | [100 200] | 1188 | 1.03 | 0.68 | 0.837 | -0.02 | 2.38 | 0.22 | 0.9490 | 79.21 |
| | [200 500] | 1477 | 0.99 | 0.63 | 0.780 | -0.06 | -0.67 | 0.23 | 0.9460 | 79.42 |
| | [500 2600] | 1067 | 1.04 | 0.60 | 0.767 | -0.08 | -2.07 | 0.28 | 0.8950 | 71.04 |
| Latitude (°N) | [-52 -35] | 85 | 1.36 | 0.43 | 0.809 | -0.07 | -3.67 | 0.28 | 0.7990 | 65.88 |
| | [-35 -20] | 217 | 1.71 | 0.64 | 0.772 | 0.00 | 2.67 | 0.36 | 0.8300 | 77.88 |
| | [-20 0] | 67 | 2.66 | 0.90 | 0.716 | -0.13 | -2.12 | 0.43 | 0.8890 | 80.60 |
| | [0 20] | 140 | 2.94 | 1.01 | 0.764 | -0.20 | -3.89 | 0.54 | 0.8670 | 70.00 |
| | [20 35] | 1226 | 1.59 | 0.82 | 0.729 | -0.16 | -5.22 | 0.39 | 0.9040 | 62.56 |



| [35 50] | 3023 | 0.93 | 0.45 | 0.771 | -0.05 | -0.05 | 0.19 | 0.9150 | 77.97 |
| [50 60] | 1467 | 0.76 | 0.35 | 0.860 | 0.00 | 1.89 | 0.13 | 0.9270 | 86.09 |
| [60 76] | 100 | 0.64 | 0.28 | 0.838 | 0.00 | 2.97 | 0.11 | 0.9250 | 91.00 |

## 5 Application of PWV product in Qinghai-Tibet Plateau


The Qinghai-Tibet Plateau (QTP) plays an important role in regional weather and climate, especially for East Asia. As
we all know, water vapor can significantly affect climate change, radiation balance and hydrological cycle. Thus, studying
for the atmospheric water vapor distribution over QTP is useful to understand the influence of QTP on the weather and
climate. However, the ground-based observations of PWV are sparse and unevenly distributed, so it is difficult to investigate
the distribution of PWV over QTP with ground-based observations. Satellite-based measurement has been widely used in the
analysis on the distribution of PWV over QTP owing to its advantage of large area coverage. In this section, the seasonal
variation and distribution of PWV over TP are analyzed with the MERSI-II measurements from September 2018 to June
2021.

Figure 9 (a, b) show the distribution of PWV over QTP during the warm (April to September) and cold (October to
March) seasons. The PWV shows a distribution consistent with altitude, and there is high PWV in low altitude but small
PWV in high altitude. The large PWV is centered in the Bay of Bengal, with values above 4.0 cm and 2.0 cm in warm and
cold seasons, respectively. The small PWV is mainly located over the western part of Tibet and it is more significant in cold
season with a large area having the PWV less than 0.5 cm. Around the Brahmaputra River, which is a precipitation center
over QTP, an obvious water vapor transport path lies along the Brahmaputra Grand Canyon. The water vapor from the Bay
of Bengal region is transported into QTP through this path, making a higher PWV in this area. Therefore, a comparison
between the two stations of Motuo and Shimian is analyzed to shed light on the role of the Brahmaputra Grand Canyon in
the transport of water vapor. The two stations, Motuo and Shimian, are situated at similar latitudes but different longitudes
(Figure 9c). It is noted that the altitude of Motuo station is 1279.0 m, higher than the altitude of Shimian station (875.1 m).
Figure 9d shows the annual variation of PWV at both sites represented as box diagrams, which are defined as follows:
bottom and top of boxes denote the 25th and 75th percentiles with the horizontal lines inside the box being the median. The
dotted lines represent the range of the adjacent value, which is the most extreme value that is not an outlier, and the outliers
are marked by crosses (Zhang et al., 2020). As we have discussed above, it is reasonable that large PWV should be found in
a low altitude generally. However, the trends of PWV at the two sites are similar, and there are nearly identical PWV mean
values for both sites. Besides, the annual variation of PWV shows that the PWV of Motuo site is obviously higher than that
of Shimian in July, which means that the PWV transport of the Brahmaputra Grand Canyon is more significant at this
moment.

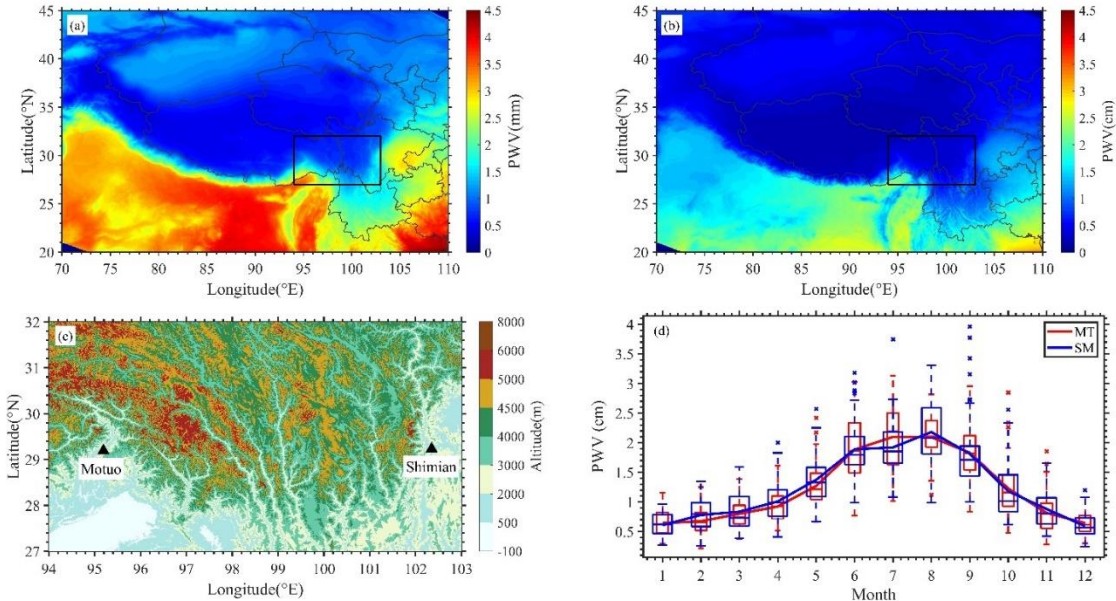


Figure 9 The distribution of PWV over QTP in (a) warm and (b) cold seasons. (c) Map illustrating the Motuo and Shimian
stations. (d) Statistics of PWV at Motuo (red line) and Shimian (blue line) stations represented as box diagrams. Bottom and
top of boxes denote the 25th and 75th percentiles, with the horizontal lines inside the box being the median value; the dotted
lines represent the range of the adjacent value, which is the most extreme value that is not an outlier; the outliers are marked
by crosses. The lines are the mean seasonal PWV.

## 6 Summary and Conclusions

In this paper, we have evaluated the global PWV product derived from FY-3D/MERSI-II by comparison to the PWV
from 626 IGRA stations, with 54214 matchup points during the period from September 2018 to June 2021. There is a good
agreement between the average PWVs from FY-3D/MERSI-II and IGRA, but the FY-3D/MERSI-II PWV is slightly
underestimated. The averaged PWV from MERSI-II and IGRA both are presented as the distribution opposite to latitude,
and generally featured with one low center over the east of Russia and the northeast of China, and two high PWV areas
concentrated in the surrounding areas of the Bay of Bengal and the central part of South America.

Overall, PWVs derived from MERSI-II and IGRA have a good agreement with the CC value of 0.9400. The values of
MB and MRB are -0.09 cm and -1.90%, respectively, while the RMSE is 0.31 cm with a satisfactory EE value of 75.36%.
Histograms of MB and MRB indicate that the values of MB and MRB both approach zero, however, the distribution patterns
are left-skewed and right-skewed, respectively. The peak values of MB and MRB are 0.00 cm and -2.38%, with STDs are



0.25 cm and 16.8%. For all sites, the MB value is low and 80% of the sites have the values between -0.28 cm and 0.05 cm.
In the west and south of Asia, the MERSI-II PWV is obviously underestimated. However, the overestimated PWV are
mostly distributed in the surrounding areas of the Black Sea and the middle part of South America. Large MRB value mostly
lies in eastern Russia, northeastern China, and central South America. 90% of all sites have low RMSE values below 0.49
cm and CC values above 0.8213.

In winter, the values of MB and RMSE are the lowest, being -0.04 cm and 0.27 cm, respectively. For MRB, it has

maximum value in summer but minimum value in spring, and apart from that, the MRB is positive in winter due to the small
PWV with a high positive MB in winter. The CC value is larger than 0.9080 in all four seasons and the EE value varies from
73.10% to 77.47%. There is a significant monthly variation in the evaluation of MERSI-II PWV product. The peak MBs are
in February and July over the Southern and Northern Hemisphere, respectively. The largest RMSE is 0.40 cm in December
in Southern Hemisphere. Besides, there is a better correlation between PWVs derived from MERSI-II and IGRA in Southern
Hemisphere, and all CC values are larger than 0.9070 except in July. The EE values during all months are larger than 66%,
indicating that there is a satisfactory coherence between the PWVs from MERSI-II and IGRA.

In addition, the influence factors on the evaluation are also discussed. First of all, the influence of temporal discrepancy

between the passing time of FY-3D and the release time of radiosonde is analyzed. There are some differences within
different temporal intervals. MRB has the largest value with temporal discrepancy of 0-1 h and the minimum value is found
when the temporal discrepancy is 1-2 h. EE value declines with the ascending temporal discrepancy from 68.04% to 88.82%.
However, there is no noteworthy difference in MB within different temporal intervals and the MB value changes from -0.06
cm to -0.10 cm. For RMSE, the greatest value is 0.36 cm, seen at the temporal discrepancy of 4-6 h. All of CC values are
larger than 0.9320 and the best correlation is found when the temporal discrepancy is less than 1 h. Subsequently, the
evaluations within different distance intervals are presented in order to reveal the effect of distance between the footprint of
FY-3D and radiosonde sites location. The MB varies positively with the growth of the distance and the largest MB is -0.15
cm within the distance of 20-100 km. The MRB is increasing from -0.49% to -5.93% with the increasing distance. However,
the CC value is less than that in different temporal intervals, besides, there are larger MB, MRB and RMSE in the evaluation
of MERSI-II PWV with different distance discrepancy intervals, and this can be concluded as the discrepancy of distance has
more effect on the evaluation of PWV than temporal discrepancy. In general, large MB and RMSE are both distributed at the
high-altitude stations, with the values of -0.08 cm and 0.28 cm, and the STD becomes smaller with the increase of altitude.
However, the least EE value is found in high altitude sites. From the analysis of latitudinal performance of MERSI-II PWV,
the MEAN of PWV shows a distribution opposite to latitude. The largest values of MB and MRB are within 0°N -20°N and
20°N -35°N, respectively. The RMSE is less than 0.19 cm above the latitude of 35 °S, however, the RMSE has large value
around the equator with the value greater than 0.43 cm.





Finally, the PWV product derived from MERSI-II is employed to analyze the PWV distribution over QTP. In Both

warm and cold seasons, the large PWV is concentrated in the Bay of Bengal, and the values are above 4.0 cm and 2.0 cm,
respectively. As the distribution of PWV shows in clear sky condition, the water vapor transport path along the Brahmaputra
Grand Canyon is obviously with a large PWV. What's more, the comparison between the monthly variations of PWV at
Motuo and Shimian sites suggests that the two stations both enjoy the nearly identical PWV mean values. In terms of the
altitudes of the two stations, the results indicate that the Brahmaputra Grand Canyon plays a key role in the transport of
water vapor, especially in July.
**Data availability**
The MERSI-II PWV product is available from http://satellite.nsmc.org.cn/PortalSite/Data/Satellite.aspx, the IGRA data is
available from ftp://ftp.ncdc.noaa.gov/pub/data/igra, and the global AERONET data are provided at
https://aeronet.gsfc.nasa.gov. The altitude data set is provided by Geospatial Data Cloud site, Computer Network
Information Center, Chinese Academy of Sciences at http://www.gscloud.cn. The processed data are available from Zenodo
(https://doi.org/10.5281/zenodo.5105083).
**Author contributions**
Conceptualization, ZWG; data curation, WL, YY and HQX; formal analysis, ZWG, YY and XGR; writing-original draft
preparation, ZWG; writing-review and editing, ZWG and WL; supervision, XGR and HXQ; funding acquisition, XGR and
CCG. All authors have reviewed and agreed on the final version of the manuscript.
**Competing interests**
The authors declare that they have no conflict of interest.
**Acknowledgments**
This work is supported by The Second Tibetan Plateau Scientific Expedition and Research (STEP) program (Grant No.
2019QZKK0105); National Natural Science Foundation of China (NSFC) under Grant No. 41705019 and 41620104009; the
Hubei Meteorological Bureau project under Grant No. 2018Q04; and NSFC under Grant No. 91637211.We appreciate the
National Satellite Metrological Center of China Meteorological Administration (CMA) for providing the MERSI-II PWV





product, the National Climatic Data Center (NCDC) for providing IGRA data, and the principal investigators and their staff
for establishing and maintaining the AERONET  sites used in this study. The altitude data set is provided by Geospatial Data
Cloud site, Computer Network Information Center, Chinese Academy of Sciences.

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
