# Peer review of "Global Evaluation of the Precipitable Water Vapor Product from 2 MERSI-II onboard the Fengyun-3D Satellite"

_Atmospheric Measurement Techniques, 2021_

## Referee Comment (RC1)

The paper by Zhang et al. describes results from an analysis of Precipitable Water Vapor (PWV) data from the MERSI-II instrument on Fengyun-3D. The satellite data are compared to ground based radiosonde data, and a example application for the Qinghai-Tibet Plateau is described.

Although the MERSI-II PWV seems to be a good product, I think the paper needs a major revision before publication. Especially, the issues listed below need to be addressed.

**General Comments:**

1. The paper uses MERSI-II PWV data, but there is no detailed information (or reference) given about the retrieval algorithm, possible adaptations specific for MERSI-II data, any assumption made or possible external data used in the generation of the PWV product.

2. The paper presents a statistical analysis of collocated data without much physical interpretation. For example, it is not clear (or at least discussed) if observed differences between satellite and ground based data are significant in view of atmospheric variability and/or errors of the products.

3. Actually, the observed differences between the MERSI-II PWV and the radiosonde data are quite small (only a few percent). I would expect that this is within the expected precision/accuracy of the involved products, especially taking into account the large spatial and temporal variability of water vapor. Nevertheless, a lot of effort is made to (statistically) quantify the reasons for these differences, namely distances in space or time or surface altitude, although the findings are essentially as expected (e.g. better agreement for smaller distances). Therefore, this part of the paper can be significantly shortened.

4. The conclusions drawn for the Qinghai-Tibet Plateau application are not supported by the corresponding data/analysis. They are mainly based on the comparisons of two stations, which actually show a very similar seasonal variation (the data agree within their 25%/75% percentile ranges). So, how can there be any conclusion about transport be drawn? Of course, water vapor transport is affected by the terrain / surface elevation, but this is no new finding.

5. The English of the paper needs improvement. In may cases, 'the' is missing in the sentences (also in the title before 'Precipitable'); use of singular/plural is wrong, and wording/formulations are sometimes misleading, e.g., when using in the abstract 'distance discrepancy' instead of 'spatial distance'. Some of these issues are addressed in this review, but there are way too many to be listed completely.

**Specific Comments:**

1. l. 15:
   Mentioning a 'peak value of the MB' of 0.00 is misleading, especially since in the following underestimations are mentioned. It should be clarified that 'peak value' refers to the histogram analysis.

2. Introduction:
   In the Introduction only NIR satellite water vapor measurements are addressed. PWV is measured from satellites at various spectral regimes (VIS, NIR, SWIR, TIR, MW, ...); especially the MW measurements have a very long heritage. This should at least be mentioned.

3. l. 30:
   'upward radiance over the view of satellite' – unclear formulation. Probably you mean that the radiance measured by the satellite instrument is affected by water vapor. Please clarify.

4. l. 30:
'For MERIS, the PWV retrieval algorithm employing the ratio of top of atmosphere (TOA) radiance at one water vapor absorption channel (900 nm) to TOA radiance at 885 nm, which is outside water vapor absorption region (Bennartz and Fischer, 2001).'

This is not even a real sentence - please reformulate.

5. Section 2.1:
This section lacks a description (or a proper reference to) the MERSI-II PWV retrieval algorithm. Some information is given in the referenced paper from Yang et al. (2019), but this is not clear from how the paper is referenced now, and the contained information is very limited. Especially, information should be given on assumptions made in the retrieval, any external data used in the retrieval, consideration of surface elevation, expected (random) error of the product, handling of aerosols, etc. — all information which actually defines the product. Furthermore, some instrument details should be given, e.g. swath width and spatial/temporal coverage.

6. l.121:
The radiosonde product is defined as surface-to-500-hPa PWV - this means that it does not include some percents of the total column. Has this been considered in the comparisons? This may be relevant as observed differences are also in the order of a few percent.

7. Section 3.1:
This section could possibly be shortened as all statistical indicators except for EE are quite common. However, the definition of EE is not fully clear. In the text, EE is described as a limit in percent whereas eq. 5 results in absolute values. Furthermore, eq. 5 includes an offset of 0.05 which needs to be explained - is this specific for IGRA data and derived from some validation activities? Please clarify.

8. l. 164–165:
'In processing, only the point that has matching data in each interval is selected for the comparison reliability.'

This sentence is unclear, please clarify / reformulate.

9. Fig. 1:
Why is the number N of collocations the same for all time intervals?

10. l. 179:
Please justify the 0.25 cm limit for STD used here.

11. l. 190:
'the closest PWV retrieval ... is selected'

Do you mean spatially closest?

12. Section 4.1:
Instead of Fig. 3, this section should include a global map of MERSI-II PWV data (e.g. daily or monthly, depending on coverage) on which the global water vapor distribution is discussed. Doing this solely based on the collocated data (Fig. 3) is not appropriate, because as shown in Fig. 2 the sampling of each site is different. The average at one station may contain e.g. different seasons for each station and is not representative for a global value. Inclusion of a full daily or monthly map would also show the coverage of the MERSI-II data.

13. l. 219–221:
'there are some individual points with the unnormal MB and MRB. Therefore, the top 1% and bottom 1% of MB and MRB are not present in the histogram in order to show an intuitive acknowledge of distributions of MB and MRB.'

This is unclear. What do you mean with 'unnormal' and 'intuitive acknowledge'? Do you mean that there is an outlier filter applied here? Is this part of the MERSI-II product (if not, why do you need it)? Is this filter also applied for other analyses in the paper?

14. Fig. 4 and related text:
The histograms given in Fig. 4 are produced using a certain bin width, which should be specified. This bin width should also be considered when specifying values in the text (no. of digits). For example, you cannot derive an MRB value of -2.38% (2 digits accuracy) if the bins are e.g. 1%.

15. l. 228–230:
'this result is comparable to the accuracy of MODIS NIR PWV product, which is compared with MWR PWV and with a 5%-10% error range.'

To which MERSI-II/MODIS values do you refer here (MRB, STD)? Note that the accuracy of the MODIS product is not the same as an error.

16. Fig. 5 and related text:
The percentage values given in the text cannot be inferred from Fig. 5, they should be taken from the histograms in Fig. 4. Actually, it seems that the given numbers are not in line with Fig. 4, e.g.:

l. 238: 'About 80% of all sites have negative MB values'

From Fig. 4b, it should be 60%. Please check/harmonize.

In general, because the sampling at each station is different, it is unclear how representative Fig. 5 and the conclusions on spatial distribution of deviations etc. are. This needs to be discussed.

Please also discuss some potential (physical) reasons for the spatial distribution of differences in addition to sampling. Are these due to possible limitations of the measurements (e.g. low signal, high noise) or local features (e.g. surface reflectance, aerosols, ...).

17. Section 4.2:
This part could possibly be shortened. Is, for example, Fig. 6 (seasonal results) needed in addition to Table 1 (monthly results)?

As this section is about temporal variations it would be good to show at least one example for a full time series at a station in a plot.

It should also be justified why the results are considered to be significant noting that e.g. differences in MB for different months in Table 1 are small.

18. l 271–272:
'with a high positive MB in winter'

Actually, the MB given in Fig. 6d is negative and small.

19. Section 4.3:
As already mentioned in the General Comments, it should be justified why a detailed analysis of influence factors is required at all.

The observed deviations are small, they have to be put in relation to the (currently not specified error) of the product. In view of natural variability and sampling issues, is the analysis accurate enough to separate (sub-)percentage effects?

Qualitatively, the results shown in this section are in most cases as expected – the closer the correlative date in time and space the better the agreement. One exception is that MRB is smallest for a temporal distance of 1–2 h instead of 0–1 h, but there is no explanation given for this.

Actually, an analysis of temporal effects was already shown in section 3.2 using AERONET data with the conclusion that 6 h differences are sufficient. So, why investigate this again?

20. l. 341–343:
   'there is no height correction that can be used to eliminate false signals...'

   Which height is assumed in the algorithm? Is varying surface elevation considered? This is an example why more information about the algorithm is needed.

21. l. 347–349:
   How are the 'hazy conditions' identified?

   'the influence of haze is hardly corrected completely in the MERSI-II PWV retrieval algorithm'

   Does this mean there is a correction or not? If yes, how does it work?

22. Table 2 and related text:
   Noting the different sampling of the stations and their distribution with latitude it should be justified why the presented values are significant.

   How much do the results depend on the selected month?

   Regarding altitude dependence, what is the vertical sensitivity of the MERSI-II measurements?

23. Section 5:
   As mentioned in the General Comments, it is not clear how transports effects can be identified solely from time series from two single stations, especially since these time series are almost identical. The values at the stations for July agree within the percentiles - why is this a significant difference? For the identification of transport one would at least need e.g. a time series at the source region (Bay of Bengal) which shows a peak during a certain month which is then observed with some delay at one of the stations. The selection of stations should also be more motivated - why exactly these? What is the role of surface elevation / surrounding terrain?

   In general, it is very difficult to follow the argumentation in this section.

24. l. 388:
   'the trends of PWV at the two sites are similar'

   There are no trends shown in Fig. 9. Do you mean 'seasonal variation'?

25. Fig. 9d and related text:
   Please confirm that the PWV data shown in the time series are from MERSI-II and not radiosondes.

**Technical Corrections:**

1. In most cases, too many digits are given for values in the text. Only significant digits (in relation to the uncertainty of the products and the analysis methods) should be given. Two digits should be usually sufficient, in case of percentage values even less.

2. Color bars of figures:
   Figures 1–8 use a discrete color bar which contains quite similar colors at the lower and higher end (light pink, orange) which can be misleading. Furthermore, the steps between colors are in most cases not equidistant, which makes a visual interpretation difficult, especially for the map plots. I suggest to use for all figures a color bar with equidistant steps and colors e.g. similar to the ones in Fig. 9, or maybe even a conceptually uniform color map.

3. Fig. 1 (and actually all scatter plots):
   '=EE:' although explained in the text, this notation (with a '=' in the variable name) is a bit misleading. I suggest to rename this.

4. Fig. 1:
   Remove symbols on top of panel (a).

5. Figs. 3 and 5 (and possibly Fig. 2):
   As these show global maps I suggest to remove the labels on the latitude and longitude grid.

6. Fig. 5:
   Please describe panels a–d in the caption and label in each figure the quantity (MB, ...) shown.

7. Figs. 7 and 8:
   Suggestion: These figures are not really needed - their results could be placed in a table instead.

8. l. 334:
   'observations in April are selected in the comparison rather than the annual mean value (MEAN)'

   As I understand, MEAN is not the annual mean value but the mean over all values in April.

9. Table 2:
   Remove/adapt line breaks in left column / top line.

10. l. 366–367:
    'As we all know, water vapor can significantly affect climate change, radiation balance and hydrological cycle.'

    This sentence is essentially identical to one in the introduction and can be deleted.

11. Fig. 9:
    Please include/mark the Brahmaputra River and the Brahmaputra Grand Canyon in the maps. This would help a lot to follow the argumentation.

---

## Referee Comment (RC2)

The paper entitled 'Evaluation and Application of Precipitable Water Vapor Product from MERSI-II onboard the Fengyun-3D Satellite by Wengang Zhang'. The obtained results are interesting to know more about the satellite based observations uncertainties on temporal, spatial (distance) & altitude based and try to improve them. The conventional observations are expensive and limited (twice in a day). Space based PWV have inherent uncertainties and need to be validated time to time basis before using its with corrected bias confidence. In that respect the present study have a high potential for publication after incorporation of the comments/suggestions as given below:

Line -106-7: Details about retrieval algorithm near-infrared Precipitable water vapor product from MERSI-II are missing and also give references.

Line-108: Which method was used to identify cloudless pixels?

Section2.2    : You have used Radiosonde & AERONET data as base for comparison with MERSI-II . But the Radiosonde & AERONET based data also associated with errors. Explain the possible sources of Radiosonde & AERONET errors in your analysis.

Line 163: the consistency between the existing AERONET PWV and AERONET PWV measurements in varioustemporal discrepancy intervals from 1 h to 6 h is analyzed. I do not understand the paragraphs.

Line 177-178: For the MERSI-II, the spatial resolution at nadir is 1 km × 1 km for NIR bands, which are used for the retrieval of PWV. Therefore, we use the standard deviation (STD) of a box with 9×9 pixels to eliminate the invalid PWV measurement. In operation, we set a general principle that the STD of this selected box must be less than 0.25 cm and the value of the STD dividing the minimum within the selected box must be less than 1.
Why you have set the limit of <0.25 cm? Why you have not set 1 or 2sigma STD to check the data quality.
Line 172: Figure 1 Authors should recheck the caption.

And line 189-191: In processing, all the PWV retrievals derived from MERSI-II within ±6 h of radiosonde release time are all collected and the closest PWV retrieval of MERSI-II within 100 km distanced from the IGRA site is selected and matched up with IGRA PWV.

I could not catch the match up criteria applied by authors. Explain whether any interpolation technique used to interpolate the data from 1x1 Km to 100 Km.

Line 200-208: There is a lack of discussion about meteorological / physical interpretation in cause of High & Low PWV centers. Around the tropics with latitude between 20°S and 20°N, the greatest PWV values are found with most PWV values above 2.17 cm. Lower PWV values are presented in mid-latitude, but the variability of PWV is the largest here with the values range from 0.60 cm to 2.17 cm. The PWV values in high latitudes are the lowest and most sites have the PWV values below 1.44 cm.

Line 217-218: Give reference.

Line 230-232: The radiosonde ascents drift and vertical extent will be different over different geographical domains. Similarly, the collocations matchups of clear sky pixel retrievals will vary and hence the MB and MRB values also vary latitudinal.

Line 256-257: Why low CC values that smaller than 0.8213 are predominantly concentrated around the equator. Give some reasons.

Line 267: Give references.

Line 269-270: Whether RMSE values are higher under the wet conditions [summer (JJA), autumn (SON)] than under dry conditions [spring (MAM) and winter (DJF)].

Line280-281: Give explanation regarding underestimation of MERSI-II PWV with respect to IGRA PWV for all the months in the northern as well Southern Hemisphere.

Line 286-287: Why the RMSE in the Northern Hemisphere is slightly smaller than that in the Southern Hemisphere. Give some possible reasons.

Line 333: Rephrase the sentence.

Line 348-350: the influence of haze is hardly corrected completely in the MERSI-II PWV retrieval algorithm. There is a high correlation between MERSI-II PWV and IGRA PWV, and the CC value is all above 0.8950. and the comparison of altitudes within 100-200 m presents a better performance.

Whether influence of haze correction is applied in retrieval of MERSI-II PWV? Please clarify and improve the discussion.

Line 356-367 Authors should mention values of MB and MRB.

Line 388-391: However, the trends of PWV at the two sites are similar, and there are nearly identical PWV mean values for both sites. Besides, the annual variation of PWV shows that the PWV of Motuo site is obviously higher than that of Shimian in July, which means that the PWV transport of the Brahmaputra Grand Canyon is more significant at this moment.

It is not look like trend; It should be warm and seasonal variations of PWV. In the month of July, movement of monsoon trough towards foothill of Himalaya may increase the value of PWV. Whether Shimian site is located leeward side?

Line 440:446: Finally, the PWV product derived from MERSI-II is employed to analyze the PWV distribution over QTP. In Both warm and cold seasons, the large PWV is concentrated in the Bay of Bengal, and the values are above 4.0 cm and 2.0 cm, respectively. As the distribution of PWV shows in clear sky condition, the water vapor transport path along the Brahmaputra Grand Canyon is obviously with a large PWV. What's more, the comparison between the monthly variations of PWV at Motuo and Shimian sites suggests that the two stations both enjoy the nearly identical PWV mean values. In terms of the altitudes of the two stations, the results indicate that the Brahmaputra Grand Canyon plays a key role in the transport of water vapor, especially in July. It is a simple comparison of two stations in respect of warm and cold seasonal variations of PWV. It is advised to do further case study combining the specific synoptic patterns (such as the background circulation, the thermodynamic conditions, etc.).

---

## Author Comment (AC1)

**Changes in the Revision**

(1) More details about the retrieval algorithm of MERSI-II PWV product are presented.

(2) The analysis in section 4 is improved and section 4.2 is shortened.

(3) The section 5 is removed.

(4) The discussion is improved.

(5) All figures are replaced.

(6) Some related references are cited.

(7) Some sentences are rewritten.

(8) The English of the paper has been improved.

**Responses to the Reviewer's Comments**

Thanks for the reviewer to provide very useful comments and suggestions, and please see our responses in the following:

**General Comments**

(1) There is no detailed information (or reference) given about the retrieval algorithm, possible adaptations specific for MERSI-II data, any assumption made or possible external data used in the generation of the PWV product.

Response: Thank you. The detail of the retrieval algorithm of MERSI-II PWV is presented in the revision (lines 116-156, tracked manuscript).

(2) The paper presents a statistical analysis of collocated data without much physical interpretation.

Response: Good suggestion. More physical interpretations of differences between IGRA and MERSI-II are added in the revision, and there are some possible reasons added.

(3) Actually, the observed differences between the MERSI-II PWV and the radiosonde data are quite small (only a few percent). I would expect that this is within the expected precision/accuracy of the involved products, especially taking into account the large spatial and temporal variability of water vapor. Nevertheless, a lot of effort is made to (statistically) quantify the reasons for these differences, namely distances in space or time or surface altitude, although the findings are essentially as expected (e.g. better agreement for smaller distances). Therefore, this part of the paper can be significantly shortened.

Response: Thank you. In this section, we want to quantify the influence of some factors on the evaluation of MERSI-II PWV and it is important in the application of MERSI-II PWV. However, the factors we selected are too many and the analysis can be shortened. So, we only discussed the spatial distance and the value of IGRA PWV. All analysis and conclusion have been rewritten in the revision.

(4) The conclusions drawn for the Qinghai-Tibet Plateau application are not supported by the corresponding data/analysis. They are mainly based on the comparisons of two stations, which actually show a very similar seasonal variation (the data agree within their 25%/75% percentile ranges). So, how can there be any conclusion about transport be drawn? Of course, water vapor transport is affected by the terrain / surface elevation, but this is no new finding.

Response: Thank you. In this section, we want to get the conclusion that the PWV transport of the Brahmaputra Grand Canyon is important by comparing the PWV

variation of two stations. However, according to the suggestions from the reviewers, and the discussion with co-authors, this section is removed.

(5) The English of the paper needs improvement. In may cases, 'the' is missing in the sentences (also in the title before 'Precipitable'); use of singular/plural is wrong, and wording/formulations are sometimes misleading, e.g., when using in the abstract 'distance discrepancy' instead of 'spatial distance'.

Response: Thank you. The grammar, spelling, punctuation and phrasing of the paper has been improved.

**Specific Comments:**

(1) Mentioning a 'peak value of the MB' of 0.00 is misleading, especially since in the following underestimations are mentioned. It should be clarified that 'peak value' refers to the histogram analysis.

Response: Thank you. The sentence is rewritten (lines 16-17). The histogram of all MB shows that the MB is concentrated around zero and mostly located within the range from -1.00 cm and 0.50 cm.

(2) In the Introduction only NIR satellite water vapor measurements are addressed. PWV is measured from satellites at various spectral regimes (VIS, NIR, SWIR, TIR, MW, ...); especially the MW measurements have a very long heritage. This should at least be mentioned.

Response: Good suggestion. The information of satellite-based retrieval of PWV is added in the revision (lines 60-62). Over the past few decades, the satellite-based PWV retrieval algorithms are developed with the observations from different sensors, which mainly can be divided into four types according to the spectral region: (1) visible (VIS), (2) near-infrared (NIR), (3) thermal infrared (TIR), and (4) microwave (MW).

(3) 'upward radiance over the view of satellite' – unclear formulation. Probably you mean that the radiance measured by the satellite instrument is affected by water vapor. Please clarify.

Response: Thank you. We actually want to express the influence of water vapor in the satellite measurements and the sentence has been rewritten in the revision (lines 37-38). Furthermore, water vapor can also influence the atmospheric transmittance and the upward radiance measured by the satellite sensor.

(4) l. 30. This is not even a real sentence - please reformulate.

Response: Thank you. We have rewritten this expression in the revision (lines 71-73). For MERIS, the PWV retrieval algorithm employed the ratio of top of atmosphere (TOA) radiance at one water vapor absorption channel (around 900 nm) to the TOA

radiance at the atmospheric window channel such as 885 nm (Bennartz and Fischer, 2001).

(5) Section 2.1 lacks a description (or a proper reference to) the MERSI-II PWV retrieval algorithm.

Response: Thank you. The detail of the retrieval algorithm of MERSI-II PWV is presented in the revision (lines 116-156, tracked manuscript).

(6) The radiosonde product is defined as surface-to-500-hPa PWV - this means that it does not include some percents of the total column. Has this been considered in the comparisons? This may be relevant as observed differences are also in the order of a few percent.

Response: Good suggestion. We consider the dry bias for the radiosonde PWV, and the related citation is added in the revision (lines 168-171). As discussed by Turner et al. (2003), the PWV obtained from radiosonde has an approximate 5% dry bias compared to that derived from the MWR. Therefore, there is an underestimation of PWV evaluation for taking the IGRA-derived PWV as a reference, and the bias found in tropical areas is ~9% (Zhang et al. 2018).

(7) Section 3.1 could possibly be shortened as all statistical indicators except for EE are quite common. However, the definition of EE is not fully clear. In the text, EE is described as a limit in percent whereas eq. 5 results in absolute values. Furthermore, eq. 5 includes an offset of 0.05 which needs to be explained - is this specific for IGRA data and derived from some validation activities? Please clarify.

Response: Thank you. This section is shortened and the EE is deleted in the revision.

(8) 'In processing, only the point that has matching data in each interval is selected for the comparison reliability.' This sentence is unclear, please clarify / reformulate.

Response: Thank you. In processing, only the existing AERONET PWV, which has the matching averaged AERONET PWV in each temporal discrepancy interval, is selected for the determination of the temporal collocation window. This is presented in the revision.

(9) Why is the number N of collocations the same for all time intervals?

Response: Thank you. This can be explained by the response of above comment (8). Only the AERONET PWV is used in the comparison when it has the matching averaged PWV in each temporal interval, that is, if there is no matched PWV in any interval, the data is discarded. Therefore, there is the same number of collocations for all the temporal discrepancy intervals. And This is expressed in the revision (lines 216-219).

(10) l. 179. Please justify the 0.25 cm limit for STD used here.

Response: Thank you. In the processing of satellite data, we hope to eliminated the PWV retrieval with a large variation in the selected 9×9 box. But according to the comments from the reviewers, we are not using this criterion anymore and the data

are recalculated.

(11) 'the closest PWV retrieval ... is selected'. Do you mean spatially closest?

Response: Thank you. Yes, the spatial closest PWV retrieval within 50 km away from the IGRA site is selected. It has been discussed in the revision (lines 247-248).

(12) Instead of Fig. 3, section 4.1 should include a global map of MERSI-II PWV data (e.g. daily or monthly, depending on coverage) on which the global water vapor distribution is discussed.

Response: Good suggestion. We have presented the monthly averaged PWV and the global distribution of the PWV in the revision (lines 262-270).

(13) What do you mean with 'unnormal' and 'intuitive acknowledge'? Do you mean that there is an outlier filter applied here? Is this part of the MERSI-II product (if not, why do you need it)? Is this filter also applied for other analyses in the paper?

Response: Thank you. Actually, there are some outliers in the histogram analysis but they are mostly distributed in the tail of the histogram with a really small sample number. In order to present more detail about the distribution of the MB, only the main range of the MB is shown, and the data is not eliminated. However, in the revision, all MB are presented in the histogram without any filter applied.

(14) Fig. 4 and related text: The histograms given in Fig. 4 are produced using a certain bin width, which should be specified. This bin width should also be considered when specifying values in the text (no. of digits). For example, you cannot derive an MRB value of -2.38% (2 digits accuracy) if the bins are e.g. 1%.

Response: Good suggestion. We have rewritten the expression in the revision. For the histogram of MB, the bin width is 0.05 cm and it is described in the revision.

(15) l. 228–230: To which MERSI-II/MODIS values do you refer here (MRB, STD)? Note that the accuracy of the MODIS product is not the same as an error.

Response: Thank you. The MRB is mentioned here and the part is deleted in the revision.

(16) Fig. 5 and related text: The percentage values given in the text cannot be inferred from Fig. 5, they should be taken from the histograms in Fig. 4. Actually, it seems that the given numbers are not in line with Fig. 4, e.g.:

In general, because the sampling at each station is different, it is unclear how representative Fig. 5 and the conclusions on spatial distribution of deviations etc. are. This needs to be discussed.

Please also discuss some potential (physical) reasons for the spatial distribution of differences in addition to sampling.

Response: Good suggestion. The percentage value can be inferred from Fig. 5 and this is because we divide the bias value into normalized value, that is, each color represents 10% of all sites. It should be considered the influence of the difference of sample number at each station, and we add the discussion in the manuscript (lines

322-325, 347-348). We also describe the potential (physical) reasons for the spatial distribution of differences and the discussion is also presented in the revision.

(17) Section 4.2 could possibly be shortened. Is, for example, Fig. 6 (seasonal results) needed in addition to Table 1 (monthly results)?

As this section is about temporal variations it would be good to show at least one example for a full time series at a station in a plot.

It should also be justified why the results are considered to be significant noting that e.g. differences in MB for different months in Table 1 are small.

Response: Thank you. This section is shortened in the revision, and only the seasonal analysis is presented. Actually, this section is focused on the annual performance of MERSI-II PWV rather than the temporal variation, and the title of this section is rewritten in the revision. Furthermore, the results for different months are removed.

(18) l. 271–272: 'with a high positive MB in winter'. Actually, the MB given in Fig. 6d is negative and small.

Response: Thank you. This sentence is rewritten. For the largest MRB during winter in the Northern Hemisphere, it is may be related to the points with a small PWV value but a high positive MB, because we firstly calculate the MRB for each match-up and then average all MRB values in seasons (lines 368-370).

(19) Section 4.3: As already mentioned in the General Comments, it should be justified why a detailed analysis of influence factors is required at all.

Response: Thank you. We want to quantify the influence of some factors on the evaluation of MERSI-II PWV and it is important in the quantitative application of MERSI-II PWV. In this section, the influence factors, such as the value of IGRA PWV, the spatial distance between footprint of satellite and IGRA station, are all explored in order to quantify their effects on the evaluation of MERSI-II PWV (lines 411-412).

(20) l. 341–343: 'there is no height correction that can be used to eliminate false signals...'. Which height is assumed in the algorithm? Is varying surface elevation considered? This is an example why more information about the algorithm is needed.

Response: Thank you. In the retrieval algorithm, the varying of surface elevation is considered but there is no height correction in the matching between MERSI-II and IGRA. And this part is deleted in the revision.

(21) l. 347–349: How are the 'hazy conditions' identified? 'the influence of haze is hardly corrected completely in the MERSI-II PWV retrieval algorithm'. Does this mean there is a correction or not? If yes, how does it work?

Response: Thank you. There is only the cloud detection in the retrieval with the cloud mask product. Therefore, the hazy with a low optical depth is hardly detected. There is no correction in the MERSI-II PWV retrieval algorithm, and this should be explored in the future. (lines 592-593).

(22) Table 2 and related text: Noting the different sampling of the stations and their

distribution with latitude it should be justified why the presented values are significant.

How much do the results depend on the selected month?

Regarding altitude dependence, what is the vertical sensitivity of the MERSI-II measurements?

Response: Good suggestion. The limitation of MWR-derived CLWC under heavy rains has been indicated in the revision.

(23) In general, it is very difficult to follow the argumentation in section 5.

Response: Thank you. The section 5 is removed in the revision.

(24) l. 388: 'the trends of PWV at the two sites are similar'. There are no trends shown in Fig. 9. Do you mean 'seasonal variation'?

Response: Thank you. The section 5 is removed in the revision.

(25) Fig. 9d and related text: Please confirm that the PWV data shown in the time series are from MERSI-II and not radiosondes.

Response: Thank you. The section 5 is removed in the revision.

**Technical Corrections:**

(1) In most cases, too many digits are given for values in the text. Only significant digits (in relation to the uncertainty of the products and the analysis methods) should be given. Two digits should be usually sufficient, in case of percentage values even less.

Response: Good suggestion. The MB and RMSE are with two digits, the MRB is with one digit and the CC value is with three digits in the revision.

(2) Color bars of figures: Figures 1–8 use a discrete color bar which contains quite similar colors at the lower and higher end (light pink, orange) which can be misleading. Furthermore, the steps between colors are in most cases not equidistant, which makes a visual interpretation difficult, especially for the map plots. I suggest to use for all figures a color bar and colors e.g. similar to the ones in Fig. 9, or maybe even a conceptually uniform color map.

Response: Good suggestion. The color bar is changed in all figures. We have tried to plot all figures with equidistant steps; however, the presentation is unclearly because there is a lot of sites with the same color as shown in the below figure.

[Figure]

Besides, there is the color even unused in the presentation of this figure. Therefore, the steps between colors are not equidistant and we emphasize this in the revision (lines 318-320).

(3) Fig. 1 (and actually all scatter plots): '=EE:' although explained in the text, this notation (with a '=' in the variable name) is a bit misleading. I suggest to rename this.

Response: Thank you. We have deleted the EE description in the revision.

(4) Fig. 1: Remove symbols on top of panel (a).

Response: Thank you. The symbol is removed.

(5) Figs. 3 and 5 (and possibly Fig. 2): As these show global maps I suggest to remove the labels on the latitude and longitude grid.

Response: Good suggestion. The labels on the latitude and longitude grid is removed in the revision.

(6) Fig. 5: Please describe panels a–d in the caption and label in each figure the quantity (MB, ...) shown.

Response: Thank you. We have added the description of Fig. 5.

(7) Figs. 7 and 8: Suggestion: These figures are not really needed - their results could be placed in a table instead.

Response: Good suggestion. The results are presented in Table 3 in the revision.

(8) l. 334: As I understand, MEAN is not the annual mean value but the mean over all values in April.

Response: Thank you. Yes, MEAN is the mean over all values in April. And this part is removed in the revision.

(9) Table 2: Remove/adapt line breaks in left column / top line.

Response: Thank you. It is improved in the revision.

(10) l. 366–367: 'As we all know, water vapor can significantly affect climate change, radiation balance and hydrological cycle.' This sentence is essentially identical to one

in the introduction and can be deleted.

Response: Thank you. This sentence is deleted in the revision.

(11) Fig. 9: Please include/mark the Brahmaputra River and the Brahmaputra Grand Canyon in the maps. This would help a lot to follow the argumentation.

Response: Thank you. This section is deleted in the revision.

---

## Author Comment (AC2)

**Changes in the Revision**

(1) More details about the retrieval algorithm of MERSI-II PWV product are presented.

(2) The analysis in section 4 is improved and section 4.2 is shortened.

(3) The section 5 is removed.

(4) The discussion is improved.

(5) All figures are replaced.

(6) Some related references are cited.

(7) Some sentences are rewritten.

(8) The English of the paper has been improved.

**Responses to the Reviewer's Comments**

Thanks for the reviewer to provide very useful comments and suggestions, and please see our responses in the following:

(1) Line -106-7: Details about retrieval algorithm near-infrared Precipitable water vapor product from MERSI-II are missing and also give references.

Response: Thank you. The detail of the retrieval algorithm of MERSI-II PWV is presented in the revision (lines 116-156, tracked manuscript).

(2) Which method was used to identify cloudless pixels?

Response: Thank you. the cloud mask (CLM) product of MERSI-II is used for the selection of cloudless pixels (lines 148-149).

(3) Section2.2: You have used Radiosonde & AERONET data as base for comparison with MERSI-II. But the Radiosonde & AERONET based data also associated with errors. Explain the possible sources of Radiosonde & AERONET errors in your analysis.

Response: Good suggestion. We consider the dry bias for the radiosonde PWV, and the related citation is added in the revision (lines 168-171). As discussed by Turner et al. (2003), the PWV obtained from radiosonde has an approximate 5% dry bias compared to that derived from the MWR. Therefore, there is an underestimation of PWV evaluation for taking the IGRA-derived PWV as a reference, and the bias found in tropical areas is ~9% (Zhang et al. 2018). The bias of AERONET is explored in the revision (lines182-184).

(4) Line 163: the consistency between the existing AERONET PWV and AERONET PWV measurements in various temporal discrepancy intervals from 1 h to 6 h is analyzed. I do not understand the paragraphs.

Response: Thank you. The paragraphs are rephrased in the revision (lines 214-216). The consistencies between the existing AERONET PWV and the temporal averaged AERONET PWV in various temporal discrepancy intervals from 1 h to 6 h with a step of 1 h, that is, 0–1 h, 1–2 h, etc., are analyzed respectively.

(5) Line 177-178: For the MERSI-II, the spatial resolution at nadir is 1 km × 1 km for NIR bands, which are used for the retrieval of PWV. Therefore, we use the standard deviation (STD) of a box with 9×9 pixels to eliminate the invalid PWV measurement. In operation, we set a general principle that the STD of this selected box must be less than 0.25 cm and the value of the STD dividing the minimum within the selected box must be less than 1. Why you have set the limit of <0.25 cm? Why you have not set 1 or 2sigma STD to check the data quality.

Response: Thank you. In the processing of satellite data, we hope to eliminate the

PWV retrieval with a large variation in the selected 9×9 box. But according to the comments from the reviewers, we are not using this criterion anymore and the data are recalculated.

(6) Line 172: Figure 1 Authors should recheck the caption.

Response: Thank you. We have replaced Figure 1 and the caption is improved.

(7) And line 189-191: In processing, all the PWV retrievals derived from MERSI-II within ±6 h of radiosonde release time are all collected and the closest PWV retrieval of MERSI-II within 100 km distanced from the IGRA site is selected and matched up with IGRA PWV. I could not catch the match up criteria applied by authors. Explain whether any interpolation technique used to interpolate the data from 1x1 Km to 100 Km.

Response: Thank you. There is no interpolation technique used here. The window of spatial distance is 100 km (however, it is replaced by 50 km in the revision), and the distance between the pixel of MERSI-II and the location of the radiosonde site can be calculated. And 1×1 km is the spatial resolution of the pixel. We have rephrased this in the revision (lines 248-250).

(8) Line 200-208: There is a lack of discussion about meteorological/physical interpretation in cause of High & Low PWV centers.

Response: Thank you. The interpretations in cause of High & Low PWV centers are presented in the revision (lines 265-270).

(9) Line 217-218: Give reference.

Response: Thank you. We have rephrased this sentence (lines 287-290).

(10) Line 230-232: The radiosonde ascents drift and vertical extent will be different over different geographical domains. Similarly, the collocations matchups of clear sky pixel retrievals will vary and hence the MB and MRB values also vary latitudinal.

Response: Thank you for your good suggestion. This part is deleted in the revision and the radiosonde ascents drift and vertical extent are discussed in the manuscript (lines 232-233, 299-301).

(11) Line 256-257: Why low CC values that smaller than 0.8213 are predominantly concentrated around the equator. Give some reasons.

Response: Thank you. It has been discussed in the revision (lines 343-348). There are large biases but small CC values over the equator, and that is possibly due to the following: 1) large residual IGRA PWV above 500 hPa (Boukabara et al., 2010); 2) high content and variation of PWV (Chen and Liu, 2016); 3) the covered surface with the reflectance does not linearly correlate with the wavelength (Gao and Kaufman, 2003); 4) a small number of samples. In addition, the temporal discrepancy can also lead to bias because the discrepancy in the equatorial region is slightly larger than in other regions overall. As discussed by Alraddawi et al (2018), for MODIS PWV, there are also noteworthy latitudinal decreases for MB, MRB and RMSE.

(12) Line 267: Give references.

Response: Thank you for the good suggestion. We have rewritten this expression in the revision (lines 364-367). With abundant water vapor in summer, clouds are easily to form, however, thin clouds are difficult to be measured by satellite due to their low optical depth (Solbrig, 2009; Naumann and Kiemle, 2020). Therefore, the higher underestimation of PWV in summer is probably triggered by the weakened or covered radiation signal under the thin cloud.

(13) Line 269-270: Whether RMSE values are higher under the wet conditions [summer (JJA), autumn (SON)] than under dry conditions [spring (MAM) and winter (DJF)].

Response: Thank you. Yes, there is a larger RMSE under the wet conditions than that under dry condition.

(14) Line280-281: Give explanation regarding underestimation of MERSI-II PWV with respect to IGRA PWV for all the months in the northern as well Southern Hemisphere.

Response: Thank you. This part is deleted in the revision.

(15) Line 286-287: Why the RMSE in the Northern Hemisphere is slightly smaller than that in the Southern Hemisphere. Give some possible reasons.

Response: Thank you. We have added explanations in the revision (lines 372-375). The RMSE in the Northern Hemisphere is slightly smaller than that in the Southern Hemisphere, where the greatest RMSE value is 0.49 cm in summer. There is a large oceanic coverage in the Southern Hemisphere, with a larger mean PWV than that in the Northern Hemisphere (Chen and Liu, 2016). Thus, this is a possible reason for large RMSE in the Southern Hemisphere, considering the increasing bias of the remote sensing PWV with the larger PWV value.

(16) Line 333: Rephrase the sentence.

Response: Thank you. This part is deleted in the revision.

(17) Line 348-350: the influence of haze is hardly corrected completely in the MERSI-II PWV retrieval algorithm. There is a high correlation between MERSI-II PWV and IGRA PWV, and the CC value is all above 0.8950. and the comparison of altitudes within 100-200 m presents a better performance.

Whether influence of haze correction is applied in retrieval of MERSI-II PWV? Please clarify and improve the discussion.

Response: Thank you. There is only the cloud detection in the retrieval with the cloud mask product. Therefore, the hazy with a low optical depth is hardly detected. There is no correction in the MERSI-II PWV retrieval algorithm, and this should be explored in the future. (lines 592-593).

(18) Line 356-367 Authors should mention values of MB and MRB.

Response: Thank you. This part is deleted in the revision.

(19) Line 388-391: It is not look like trend; It should be warm and seasonal variations of PWV. In the month of July, movement of monsoon trough towards foothill of Himalaya may increase the value of PWV. Whether Shimian site is located leeward side?

Response: Thank you. This part is deleted in the revision.

(20) Line 440:446: It is a simple comparison of two stations in respect of warm and cold seasonal variations of PWV. It is advised to do further case study combining the specific synoptic patterns (such as the background circulation, the thermodynamic conditions, etc.).

Response: Thank you. This part is deleted in the revision.

---

## Author Comment (AC3)

**Changes in the Revision**

(1) More details about the retrieval algorithm of MERSI-II PWV product are presented.

(2) The analysis in section 4 is improved and section 4.2 is shortened.

(3) The section 5 is removed.

(4) The discussion is improved.

(5) All figures are replaced.

(6) Some related references are cited.

(7) Some sentences are rewritten.

(8) The English of the paper has been improved.

**Responses to the Reviewer's Comments**

Thanks for the reviewer to provide very useful comments and suggestions, and please see our responses in the following:

**General Comments**

(1) However, in my opinion section 5 is unnecessary and is not in line with the rest of the manuscript. I do not really see how it fits with the rest of the paper.

Response: Thank you. The detail of the retrieval algorithm of MERSI-II PWV is presented in the revision (lines 116-156, tracked manuscript).

(2) I recommend to seek help from a native speaker to revise the English writing.

Response: Thank you. The grammar, spelling, punctuation and phrasing of the paper has been improved.

(3) Regarding the use of IGRA PWV data, which is integrated up to 500 hPa, Zhang et al. (2018) showed that in tropical regions this can induce important dry bias (they reported a 9% error).

Response: Thank you. We consider the dry bias for the radiosonde PWV, and the related citation is added in the revision (lines 168-171). As discussed by Turner et al. (2003), the PWV obtained from radiosonde has an approximate 5% dry bias compared to that derived from the MWR. Therefore, there is an underestimation of PWV evaluation for taking the IGRA-derived PWV as a reference, and the bias found in tropical areas is ~9% (Zhang et al. 2018).

(4) MERSI-II uses solar radiation (NIR). Therefore, I think the solar zenith angle and the presence of clouds should be considered among the possible influence factors studied.

Response: Thank you and good suggestion. We have added the discussion about the effect of solar zenith angle and the presence of clouds, especially thin clous (lines 137-138, 364-365, and 590-593).

**Specific Comments:**

(1) L50: Before the start of this paragraph I miss a small introduction to the fact that satellite retrieval methods can involve bands in different ranges of wavelengths (IR, NIR,VIS, MW,...).

Response: Good suggestion. The information of satellite-based retrieval of PWV is added in the revision (lines 60-62). Over the past few decades, the satellited-based

PWV retrieval algorithms are developed with the observations from different sensors, which mainly can be divided into four types according to the spectral region: (1) visible (VIS), (2) near-infrared (NIR), (3) thermal infrared (TIR), and (4) microwave (MW).

(2) From Eq. 5 it is clear that EE 15% is used, but this is not clarified in the text. Also, I would like the authors to clarify the Eq. 5, specifically the 0.05 value added to the 0.15*PWVg.

Response: Good suggestion. We have deleted the EE description in the revision.

(3) L179. Why the STD limit is 0.25 cm and not other value?

Response: Thank you. In the processing of satellite data, we hope to eliminate the PWV retrieval with a large variation in the selected 9×9 pixel box. But according to the comments from the reviewers, we are not using this criterion anymore and the data are recalculated.

(4) L267-268. Thin clouds are claimed to be the reason for MERSI-II underestimation during Summer. Can you provide some prove or reference for this claim? If not, maybe change the sentece to a less definitive one.

Response: Thank you and good suggestion. We have rewritten this expression in the revision (lines 364-367). With abundant water vapor in summer, clouds are easily to form, however, thin clouds are difficult to be measured by satellite due to their low optical depth (Solbrig, 2009; Naumann and Kiemle, 2020). Therefore, the higher underestimation of PWV in summer is probably triggered by the weakened or covered radiation signal under the thin cloud.

(5) L333-337. I understand the logic of using one month data, but why April? This should be explained in the manuscript.

Response: Thank you. The section is removed in the revision.

(6) L343-344. Maybe this sentence should include a citation.

Response: Thank you. The section is removed in the revision.

**Technical Corrections:**

(1) L12: "626 sites" World-wide?

Response: Yes. The global evaluation is performed in this manuscript. The abstract and title is rewritten in the revision.

(2) L12-13: "both present the distribution opposite to latitude". Please rephrase this.

Response: Good suggestion. We have rephrased this sentence. The monthly averaged PWV from MERSI-II presents a decreasing distribution of PWV from the tropics to the polar regions.

(3) L15: "peak values". What is this?

Response: Thank you. This is the peak value in the histogram of the MB. The histogram of MB shows that the MB is concentrated around zero and mostly located within the range from -1.00 cm to 0.50 cm.

(4) L19: "falling" --> fall

Response: Thank you. This is changed.

(5) L20-21: please rephrase

Response: Thank you. This sentence is removed.

(6) L25: I would not use "part" here. Maybe "constituent" or "compound".

Response: Thank you and good suggestion. We have rephrased.

(7) L100: " m" should be micrometers.

Response: Thank you. This is changed in the revision.

(8) L164: "AERONET PWV and AERONET PWV". Maybe it could be something like "AERONET PWV with itself", to avoid repetition.

Response: Thank you and good suggestion. We have rephrased. The consistencies between the existing AERONET PWV and the temporal averaged AERONET PWV in various temporal discrepancy intervals from 1 h to 6 h with a step of 1 h, that is, 0–1 h, 1–2 h, etc., are analysed respectively.

(9) L311-312: Please rephrase this sentence (from "Obviously" to "interval")

Response: The section is removed in the revision.

(10) Fig. 6 and 7. I think X axis is the same (IGRA PWV), but it is labelled differently (PWV-RAOB, PWV-IGRA)

Response: The section is removed in the revision.

(11) Table 2. "Altitude" and "Latitude" got separated in two lines (Altitud-e, Latitud-e).

Response: Thank you. This section is deleted in the revision.

---

## Referee Report (RR1)

The revised paper by Zhang et al. has largely improved. All my comments to the previous version have been considered. I only have a few remaining comments to the paper, which are listed below.

**Specific Comments:**

1. It seems that all results are essentially based on a cloud-free subset of the data. This should be mentioned in abstract and conclusions.

2. l. 99:
'..., which is from MERSI ...'
This sentence is unclear. You probably mean that the same on-ground calibration as for MERSI has been done for MERSI-II; in addition, on-board (in-flight?) calibration was used. Please clarify / reformulate.

3. l. 108:
'For the NIR channels, typically with a small aerosol optical thickness that can be ignored, ...'
Do you assume that aerosol optical thickness is small? Please clarify / justify.

4. l. 114–115: 'a reflectance between 850 and 1250 nm changes approximately linearly with the wavelength'
This is a very crude approximation, especially when considering absorption. Maybe you refer here to specific bands? Please clarify / reformulate.

5. The PWV map in Fig. 3 shows very low values in the tropics compared e.g. to MODIS data for the same month. Is there a filter applied to these data, e.g. is this a cloud-free subset, or is there a saturation issue in the measurements preventing to derive high PWV amounts? Please clarify.

**Technical Corrections:**

1. Abstract:
Please add explanation of MB in abstract.

2. l. 78:
'Integrated Global Radiosonde Archive' → 'The Integrated Global Radiosonde Archive'

3. l. 185:
'radiosonde site' → 'radiosonde sites'

4. l. 272:
'the great MRB' → 'the large MRB'

5. l. 276:
'Figure 4c' → 'Figure 5c'

---

## Author Response (AR2)

**Changes in the Revision**

(1) The influence of solar zenith angle is discussed in the revision.

(2) Some related references are cited.

(3) Some sentences are rewritten.

(4) The dataset is updated.

(5) Acknowledgments is updated.

**Responses to the Reviewer1's Comments**

Thanks for the reviewer to provide very useful comments and suggestions, and please see our responses in the following:

The authors have replied to all my concerns, except the issue of the solar zenith angle. I must insist that this should be included in the study, adding SZA in Table 3, in my opinion. Other studies with GOME-2 (Roman et al. 2015), or MODIS (Vaquero-Martinez et al. 2017) showed important influence of solar zenith angle in the retrieval performance.

Response: Thank you. The influence of solar zenith angle is discussed in the revision (lines 329-334,358-365, tracked manuscript).

**Responses to the Reviewer2's Comments**

Thanks for the reviewer to provide very useful comments and suggestions, and please see our responses in the following:

**Specific Comments:**

(1) It seems that all results are essentially based on a cloud-free subset of the data. This should be mentioned in abstract and conclusions.

Response: Thank you. It is based on a cloud-free subset of the data and it is mentioned both in abstract and conclusions.

(2) l. 99: '..., which is from MERSI ...'

This sentence is unclear. You probably mean that the same on-ground calibration as for MERSI has been done for MERSI-II; in addition, on-board (in-flight?) calibration was used. Please clarify / reformulate.

Response: Good suggestion and thank you. A series of comprehensive prelaunch calibrations have been operated to ensure the high quality of the products from MERSI-Ⅱ (Xu et al., 2018), which is an advanced version of MERSI and has been significantly improved with high-precision on-board calibration and lunar calibration capabilities (Wu et al., 2020). We have rewritten this expression in the revision (lines 101-104).

(3) l. 108: 'For the NIR channels, typically with a small aerosol optical thickness that can be ignored, ...'

Do you assume that aerosol optical thickness is small? Please clarify / justify.

Response: Thank you. The sentence is rewritten in the revision (lines 112-113).

(4) l. 114–115: 'a reflectance between 850 and 1250 nm changes approximately linearly with the wavelength'

This is a very crude approximation, especially when considering absorption. Maybe you refer here to specific bands? Please clarify / reformulate.

Response: Thank you and good suggestion. We have rewritten this expression in the revision (line 119).

(5) The PWV map in Fig. 3 shows very low values in the tropics compared e.g. to MODIS data for the same month. Is there a filter applied to these data, e.g. is this a cloud-free subset, or is there a saturation issue in the measurements preventing to derive high PWV amounts? Please clarify.

Response: Thank you. This is a cloud-free subset and it is mentioned in the revision.

**Technical Corrections:**

(1) Abstract: Please add explanation of MB in abstract.

Response: Thank you. The explanation is added in the revision.

(2) l. 78: 'Integrated Global Radiosonde Archive' → 'The Integrated Global Radiosonde Archive'

Response: Thank you. It is changed.

(3) l. 185:

'radiosonde site' → 'radiosonde sites'

Response: Thank you. It is changed.

(4) l. 272:

'the great MRB' → 'the large MRB'

Response: Thank you. It is changed.

(5) l. 276:

'Figure 4c' → 'Figure 5c'

Response: Thank you. It is changed.